# Fisher-Guided Selective Forgetting:
# Mitigating The Primacy Bias in Deep Reinforcement Learning

## Abstract

Deep Reinforcement Learning (DRL) systems often tend to overfit to early experiences, a phenomenon known as the primacy bias (PB). This bias can severely hinder learning efficiency and final performance, particularly in complex environments. This paper presents a comprehensive investigation of PB through the lens of the Fisher Information Matrix (FIM). We develop a framework characterizing PB through distinct patterns in the FIM trace, identifying critical memorization and reorganization phases during learning. Building on this understanding, we propose Fisher-Guided Selective Forgetting (FGSF), a novel method that leverages the geometric structure of the parameter space to selectively modify network weights, preventing early experiences from dominating the learning process. Empirical results across DeepMind Control Suite (DMC) environments show that FGSF consistently outperforms baselines, particularly in complex tasks. We analyze the different impacts of PB on actor and critic networks, the role of replay ratios in exacerbating the effect, and the effectiveness of even simple noise injection methods. Our findings provide a deeper understanding of PB and practical mitigation strategies, offering a FIM-based geometric perspective for advancing DRL.

## 1. Introduction

Deep Reinforcement Learning (DRL) agents often suffer from a critical issue known as the primacy bias (PB), where early experiences disproportionately influence the learning process, hindering the ability to adapt to new information and achieve optimal performance (Nikishin et al., 2022). This phenomenon, related to the primacy effect in human cognition, can lead to suboptimal policies and limit generalization, presenting a significant bottleneck in the development of robust and efficient DRL systems. The core of the PB problem lies in the interplay between neural network learning dynamics and the non-stationary nature of reinforcement learning (Abbas et al., 2023; Lyle et al., 2023). Early interactions often occur during the exploration phase, when an agent's policy is far from optimal. The neural network then tends to overfit to these initial experiences, shaping its representation in a way that makes subsequent learning from novel situations more difficult (Lyle et al., 2022b). While crucial for stable off-policy learning, the replay buffer exacerbates this effect by continually reinforcing these early, potentially misleading experiences (Nikishin et al., 2022). This can lead to a "loss of plasticity," as the network loses its capacity to adapt effectively to new scenarios (Abbas et al., 2023). To mitigate the negative impact of the PB, various techniques have been proposed, ranging from periodic network resetting (Nikishin et al., 2022) to pseudo-random noise injection in the learning process (Sokar et al., 2023). However, these methods often lack a deep understanding of the phenomenon's underlying mechanisms. In this paper, we address the PB through the lens of information geometry. Specifically, we leverage the Fisher Information Matrix (FIM), a tool to characterize the local geometry of the parameter space and measure network sensitivity (Amari, 2016). Through this lens, we identify distinctive phases in the learning process that are characterized by a unique pattern in the evolution of the FIM's trace (Achille et al., 2018; Jastrzebski et al., 2021), and develop a targeted and principled mitigation strategy, Fisher-Guided Selective Forgetting (FGSF).

The contributions of this paper are therefore threefold: first, we propose a novel characterization of the PB by introducing a new method that quantifies the PB via the FIM trace evolution and its derivatives; second we introduce Fisher-Guided Selective Forgetting (FGSF) a principled mitigation strategy that relies on a geometric understanding of the PB to selectively modify network weights; third through extensive experiments across multiple environments we systematically evaluate FGSF, compare its performance against existing mitigation strategies, and analyze the impact of difference hyperparameters choices, assessing the superiority

---
[1]Anonymous Institution, Anonymous City, Anonymous Region, Anonymous Country. Correspondence to: Anonymous Author <anon.email@domain.com>.

Preliminary work. Under review by the International Conference on Machine Learning (ICML). Do not distribute.

of our proposed approach over existing alternatives.

## 2. Related Work

The primacy bias is a critical challenge in DRL, where early experiences disproportionately influence the learning process, hindering the ability of agents to adapt to new information and achieve optimal policies (Nikishin et al., 2022). This bias is particularly pronounced in off-policy learning scenarios, where early, potentially suboptimal trajectories coming in te form of state $s_t$, action $a_t$, reward $r_t$ and next state $s_{t+1}$ tuples, can dominate the replay buffer, reinforcing initial biases and leading to a "loss of plasticity" (Abbas et al., 2023; D'Oro et al., 2022). These early experiences disproportionately impact value function estimation (Lyle et al., 2022b;a; Van Hasselt et al., 2018) and can manifest in various DRL paradigms, including model-based RL (Qiao et al., 2023) and multi-task settings (Cho et al.).

Several strategies have been proposed to mitigate the PB. One of the earliest approaches involved periodic network resetting, where network parameters are reinitialized at regular intervals to prevent overfitting to initial experiences (Nikishin et al., 2022). While this approach can improve performance, it often results in abrupt performance drops upon reinitialization. Plasticity injection methods, which introduce pseudo-random noise in the learning process, aim to promote ongoing learning and adaptability, preventing the network from becoming overly specialized (Sokar et al., 2023; Nikishin et al., 2024). Self-distillation strategies also aim to preserve the plasticity of the network, by transferring the knowledge from an already trained network to a randomly initialized one (Li et al., 2024), to avoid the memorization of the first trajectories. All these approaches attempt to maintain the network's learning capacity; however, they either require a trade-off between stability and performance or lack a robust theoretical basis. Furthermore, methods have explored architecture limitations or the optimization process itself as a way to tackle this phenomenon (Obando-Ceron et al., 2024; Asadi et al., 2024; Li et al., 2023).

To understand the learning dynamics in neural networks, the Fisher Information Matrix has emerged as a valuable tool. The FIM characterizes the local geometry of the parameter space and the sensitivity of the network, with a high FIM trace magnitude during training associated with poor generalization (Jastrzebski et al., 2021). The FIM also provides insights into the loss landscape (Hochreiter & Schmidhuber, 1997) and underlies techniques for approximating the FIM (Martens & Grosse, 2015; George et al., 2018). It also has been used to design more efficient exploration strategies (Kakade, 2001) and to achieve better out-of-distribution generalization (Pascanu, 2013; Rame et al., 2022) and to build more interpretable models (Luber et al., 2023). More recently, the FIM has been leveraged in the field of machine unlearning (Xu et al., 2023), which focuses on the selective removal of information from trained models. Methods from this field use the FIM as a tool for removing the influence of specific data points or subsets of data points. Selective forgetting approaches aim to minimize the effect of unwanted data while maintaining the model's performance on other relevant data. Several techniques aim to "scrub" network weights clean of specific training data (Golatkar et al., 2020b;a) by leveraging information theoretic principles to remove information up to the final activations. The goal is to ensure that the unlearning process extends beyond just the model's weights and includes final activations as well. Such methods offer theoretical guarantees on the amount of removed information and can be implemented in practice (Ramkumar et al., 2024). This body of research provides inspiration and techniques for developing targeted methods for mitigating biases in neural networks. Our work builds on these foundations by integrating concepts from information geometry, together with the techniques from machine unlearning, to create a targeted PB mitigation strategy, by using the FIM structure to guide the selective modification of network weights in DRL.

## 3. Fisher-Guided Selective Forgetting

To effectively address the primacy bias, we introduce Fisher-Guided Selective Forgetting (FGSF), a method that combines insights from information geometry and machine unlearning. The core of our approach is based on the Fisher Information Matrix and its ability to capture the learning dynamics of neural networks.

### 3.1. The Fisher Information Matrix (FIM)

The FIM is a fundamental concept in information geometry that quantifies the amount of information a random variable carries about an unknown parameter. In the context of neural networks, the FIM provides a measure of the sensitivity of the network's output with respect to its parameters. Given a neural network with parameters $\theta$, and a probability distribution $p(x|\theta)$, the FIM, denoted as $F(\theta)$, is defined as the covariance of the score function

$$F(\theta) = \mathbb{E}_{x \sim p(x|\theta)}[\nabla_\theta \log p(x|\theta) \nabla_\theta \log p(x|\theta)^T],$$

where $\nabla_\theta \log p(x|\theta)$ is the gradient of the log-likelihood function, often referred to as the score function. This matrix describes the curvature of the loss surface around the current parameters and highlights which parameters are most sensitive to changes in the data. In practical deep learning applications, the empirical FIM is used, computed over a batch of data as follows:

$$F(\theta) \approx \frac{1}{N} \sum_i \nabla_\theta \log p(x_i|\theta) \nabla_\theta \log p(x_i|\theta)^T,$$

where $N$ is the batch size. The trace of the FIM ($\text{Tr}(F)$) is particularly relevant to our work, as it summarizes the overall sensitivity of the network's parameters.

### 3.2. Characterizing the Primacy Bias with the FIM

Our analysis reveals that the PB manifests through a characteristic two-phase pattern in the evolution of the FIM trace ($\text{Tr}(F)$) during training, as shown in the Figure 1, which represents the evolution of $\text{Tr}(F)$, the differential of $\text{Tr}(F)$, and the reward during training. This characterization of different learning periods is based on the work of Achille et al.. This pattern provides a metric to characterize and understand how early experiences disproportionately influence learning:

- **Memorization Phase:** An initial sharp increase in $\text{Tr}(F)$ during early training, characterized by a rapid exponential growth. This phase corresponds to high sensitivity to parameter updates and intensive information acquisition from initial experiences.

- **Reorganization Phase:** A subsequent sharp decrease in $\text{Tr}(F)$, despite continued improvement in task performance. This phase is characterized by a gradual decline of $\text{Tr}(F)$, settling at values lower than the peak, corresponding to reduced sensitivity to new information and a consolidation of learned patterns.

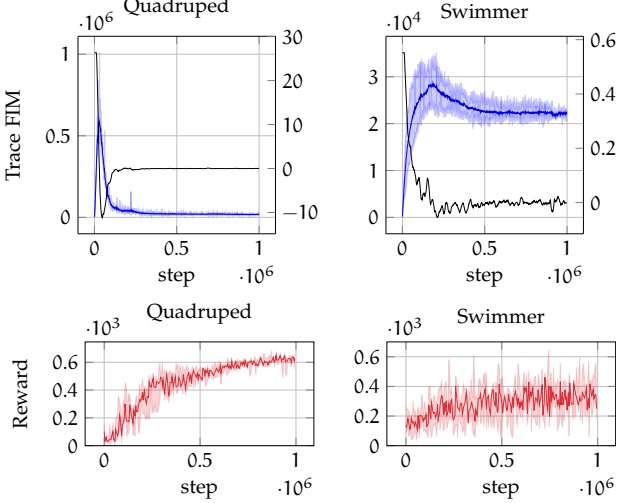

*Figure 1.* Example of the Primacy Bias characterization using the $\text{Tr}(F)$ (blue) and $\Delta\text{Tr}(F)$ (black). The graphs represent the learning dynamics on the `Quadruped` and `Swimmer` environment respectively. From our characterization, the PB is present in the `Quadruped` while it is not present in the `Swimmer`. **Best viewed in colors.**

These two phases indicate that initial experiences have a

disproportionate impact on the model, while the following phase indicates a locking of learned patterns, which results in the PB. The differential of the trace of the FIM through training ($\Delta\text{Tr}(F)$), calculated using a Savitzky-Golay filter (Candan & Inan, 2014), can highlight the transition between these two phases.

### 3.3. Selective Forgetting via the FIM

To mitigate the PB, we draw inspiration from machine unlearning and, more specifically, the selective forgetting framework introduced by Golatkar et al.. Their approach leverages the concept of a Forgetting Lagrangian:

$$L = \mathbb{E}_{S(w)}[L_{D_r}(w)] + \lambda \text{KL}(P(S(w)|D)\|P(w|D_r)),$$

where $L_{D_r}(w)$ represents the loss on the retained dataset $D_r$, while $P(w|D_r)$ represents the distribution of weights obtained after training on $D_r$ only. $S(w)$ indicates the weight scrubbing procedure, and $P(S(w)|D)$ is the resulting distribution of weights after scrubbing. Using a quadratic approximation of the loss function and assuming gradient flow optimization, the optimal scrubbing procedure can be derived as:

$$S(w) = w - B^{-1}\nabla L_{D_r}(w) + (\lambda\sigma^2)^{1/4}B^{-1/4}\epsilon,$$

where $B$ is the Hessian of the loss on the retained data, $\epsilon$ is standard Gaussian noise, and $\sigma^2$ represents the uncertainty. In practice, the Hessian $B$ is approximated with the empirical FIM (Martens & Grosse, 2015)

### 3.4. Fisher-Guided Selective Forgetting

Our final algorithm, Fisher-Guided Selective Forgetting (FGSF), tailors the theoretical framework to the DRL domain. In this context, the current batch of experiences sampled from the replay buffer is treated as the set to be retained ($D_r$), while previously encountered trajectories constitute the set to be forgotten. This interpretation aligns with our goal of preventing early experiences from dominating the learning process by periodically applying a scrubbing procedure after each standard optimization step. The scrubbing procedure is:

$$S(w) = w + (\lambda\sigma^2)^{1/4}F^{-1/4}\epsilon,$$

where $F$ is the empirical FIM, calculated from data within the current batch.

Note that, compared to the original formulation of Golatkar et al., we removed the term $B^{-1}\nabla L_{D_r}(w)$. This is justified for a couple of main reasons. First, the standard optimization process, usually based on gradient descent, already performs a similar parameter update, without the Hessian term that can be added in a second moment method for optimization as natural gradient descent (Amari, 1998; Pascanu,

**Algorithm 1** Fisher-Guided Selective Forgetting (FGSF)

1: **Input:** Current network parameters $w$, Replay Buffer $D$, Scrubbing Frequency $F$, Forget Coefficient $\lambda$
2: Initialize $t \leftarrow 0$
3: **repeat**
4:   $\{(s_t, a_t, r_t, s_{t+1}) \dots (s_k, a_k, r_k, s_{k+1})\} \sim D$
5:   Update network parameters $w$ using the DRL algorithm's update rule
6:   $t \leftarrow t + 1$
7:   **if** $t \mod F = 0$ **then**
8:     $FIM = \frac{1}{N} \sum_{i=1}^{N} \nabla_w \log p(s_i|w) \nabla_w \log p(s_i|w)^T$
9:     $\epsilon \sim \mathcal{N}(0, I)$
10:    $w \leftarrow w + (\lambda \sigma^2)^{\frac{1}{4}} FIM^{-\frac{1}{4}} \epsilon$
11:   **end if**
12: **until** Convergence

2013), hence making the gradient part redundant. Second, in contrast to the original formulation where scrubbing is performed once after training is done, our procedure is performed periodically during training, making a full gradient update unrealistic since it would drastically disrupt the optimization process.

We highlight that our proposed FGSF algorithm is compatible with any DRL algorithm that uses experience replay. For algorithms with multiple networks, such as actor-critic methods, FGSF is applied to each network independently. The scrubbing frequency and the forgetting magnitude ($\lambda$) serve as tunable parameters for balancing PB mitigation with learning stability. A fundamental interdependence exists between the scrubbing frequency and $\lambda$: more frequent scrubbing necessitates smaller $\lambda$ values to maintain stability. This relationship directly manages the trade-off between effective information removal and the preservation of learning dynamics. For the sake of simplicity, we fixed the scrubbing frequency to 10. A detailed description of the algorithm can be found in Algorithm 1.

## 4. Experimental Setup

To validate our approach and investigate the efficacy of FGSF in mitigating the PB, we conducted extensive experiments across a variety of environments and conditions. In this section, we will briefly describe the experimental setup we used in the paper.

**Environments** We evaluated our proposed method on a suite of continuous control tasks from the DeepMind Control Suite (DMC) (Tassa et al., 2018). The environments include: Basic Control Tasks: `Pendulum` and `Acrobot`. Locomotion Tasks: `Humanoid`, `Quadruped`, `Walker`, `Cheetah`, `Hopper`, and `Swimmer6`. Manipulation Tasks: `Reacher` and `Finger`.

**Algorithm and Implementation Details** For our experiments, we used the Soft Actor-Critic (SAC) algorithm (Haarnoja et al., 2018a;b) as the base DRL method. SAC was selected due to its established performance in continuous control tasks and because it is the algorithm of choice in previous work investigating the PB (Nikishin et al., 2022; Sokar et al., 2023; D'Oro et al., 2022; Li et al., 2024). We maintain the default hyperparameters of SAC, as specified in the original paper, while modifying specific parameters when explicitly studying their effects on the PB (e.g., hyperparameter study). All the experiments were performed using the Tianshou library (Weng et al., 2022) for the DRL implementation and, to compute the empirical FIM, we leveraged the NNGeometry library (George, 2021) using the Eigenvalue-corrected Kronecker-Factored Eigenbasis (EKFAC) (George et al., 2018) approximation, which provides a computationally efficient approach for estimating the FIM and is widely used in the literature.

## 5. Results

This section presents the findings of our empirical evaluation, focusing on the performance of FGSF and its impact on various aspects of DRL.

### 5.1. Comparative Analysis of FGSF

We evaluate the efficacy of FGSF by contrasting it with standard SAC implementations and periodic network reset methods, assessing performance, update magnitude (see Appendix B.1) dormant neurons (see Appendix B.2), stability, and sample efficiency to understand the advantages of our approach. Our empirical evaluation, shown in Figure 2 and summarized in Table 1 in Appendix B, reveals that FGSF exhibits a significant performance advantage, particularly in high-dimensional tasks. In the `Humanoid` environment, FGSF achieved a mean return of $150 \pm 15$, a $50\%$ improvement over baseline SAC ($95 \pm 10$), and a $25\%$ improvement over the reset method ($120 \pm 20$). Similarly, in the `Quadruped` environment, FGSF reached a final performance of $850 \pm 30$, compared to $650 \pm 25$ for baseline SAC and $780 \pm 35$ for the reset method. While the performance gap narrows in medium-complexity environments like `Walker` and `Cheetah`, with FGSF and baseline SAC reaching approximately $830 \pm 20$ in `Cheetah`, FGSF demonstrates superior sample efficiency, achieving $90\%$ of maximum performance approximately $2 \times 10^5$ steps earlier than the baseline. In simpler environments like `Pendulum` and `Reacher`, all methods attain similar final performance. Notably, FGSF shows more consistent learning without performance drops seen with the reset method. In contrast, both reset and FGSF failed to learn in the `Acrobot` environment, possibly due to hyperparameter sensitivity in this simpler environment. The `Swimmer` envi-

ronment presented minimal differences, with all approaches reaching final returns of approximately $350 \pm 30$. Across all environments, the reset method introduces significant temporary performance degradation, with sharp drops every $2 \times 10^5$ steps, unlike FGSF which provides more stable learning trajectories. FGSF consistently requires roughly $20\%$ fewer interactions than SAC in complex environments to reach performance thresholds, particularly within the initial $2 \times 10^5$ steps, indicating more efficient early-stage policy identification.

Analysis of the FIM traces, shown in Figure 3 and 8, reveals distinct patterns in how FGSF mitigates PB. In baseline SAC, both actor and critic networks show an initial sharp increase in Tr(F) during a memorization phase, reaching approximately $10^6$ for critics and $10^5$ for actors in complex environments, followed by a reorganization phase with a gradual decline. FGSF, in contrast, maintains significantly lower critic Tr(F) values (typically $10^4$-$10^5$ vs. baseline's $10^5$-$10^6$), and reduced peak magnitudes with faster stabilization in actor networks. FGSF's regulation of learning phases leads to enhanced performance, aligning with findings that reduced Tr(F) during early training correlates with improved generalization. The reset method exhibits discontinuities in the FIM trace every $2 \times 10^5$ steps, with critic networks showing faster recovery with overshoot compared to slower recovery in actor networks. In the `Humanoid` environment, baseline critic Tr(F) peaks at $2 \times 10^6$ while FGSF maintains values below $5 \times 10^5$. Based on our characterization, these FIM patterns provide evidence of FGSF's ability to mitigate the Primacy Bias.

## 5.2. Robustness Analysis

To assess the sensitivity of FGSF, we examine performance variations by exploring different noise injection coefficients ($\lambda$) and replay ratios, thereby determining the robustness of our approach concerning performance and stability.

**Hyperparameter Sensitivity** Our analysis indicates that while FGSF's effectiveness depends on the scrubbing coefficient $\lambda$, it maintains robust performance across a range of values ($5 \times 10^{-6}$ to $5 \times 10^{-8}$). This can be seen in Figure 4 and 13 and is also summarized in Table 3 in Appendix . Intermediate values, particularly $5 \times 10^{-7}$, achieve an optimal balance between learning stability and bias mitigation. Larger $\lambda$ values ($5 \times 10^{-6}$) induce aggressive forgetting and increased trajectory variability, while lower values ($5 \times 10^{-8}$) may inadequately address the PB. FIM trace analysis highlights that over-regularization (too much reduction in Tr($F$)) can disrupt the natural transition between learning phases. Surprisingly, environment complexity exhibits minimal influence on optimal $\lambda$ values, though simpler environments often show slightly better performance with lower $\lambda$. Rapid Tr($F$) oscillations indicate a need for coefficient re-

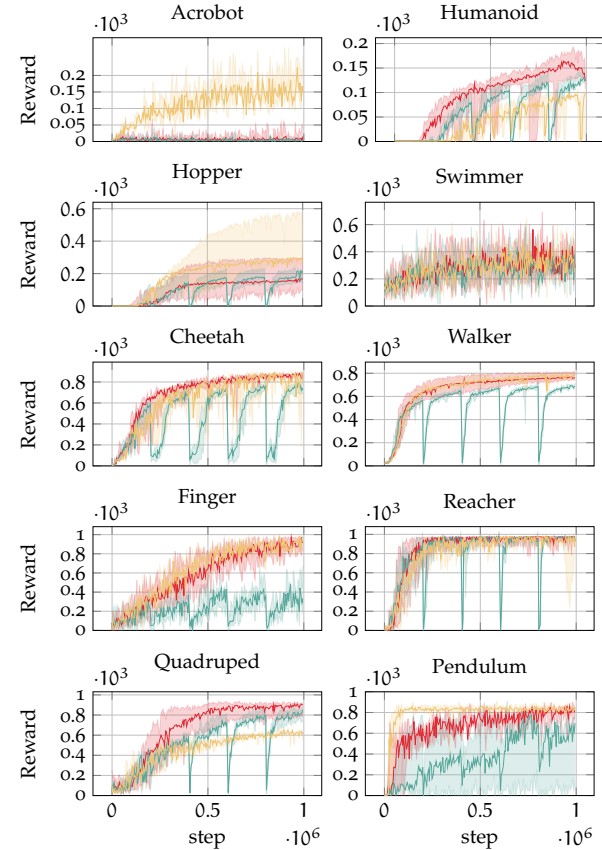

*Figure 2.* Learning curves showing episodic reward across different environments for baseline SAC (gold), reset method (teal), and FGSF (red). Shaded regions represent the minimum and maximum over 5 random seeds. **Best viewed in colors.**

duction, while inadequate post-memorization phase decline suggests insufficient $\lambda$ values. For practical implementation, we recommend an initial $\lambda$ value of $5 \times 10^{-7}$, monitoring both actor and critic FIM traces, and adjusting $\lambda$ based on observed learning stability.

**Replay Ratio** To assess FGSF's robustness in the presence of increased replay ratios, which are known to exacerbate the PB, we tested ratios of 2 and 4. As depicted in Figure 5, our results demonstrate that while higher replay ratios drastically decrease the overall performance and robustness of SAC, FGSF performs comparatively better, retaining a more robust performance. This is because when we increase the replay ratio, we are replaying the same trajectories multiple times, and, if the model got biased in the beginning of the training, these will be amplified by the replay buffer. FGSF is able to counteract this effect by making the weights less sensible to the early, biased, experiences, which leads to a higher performance with more stable learning curves. Given the relatively high FIM traces that are observed under these conditions, this suggests the need for a stronger lambda

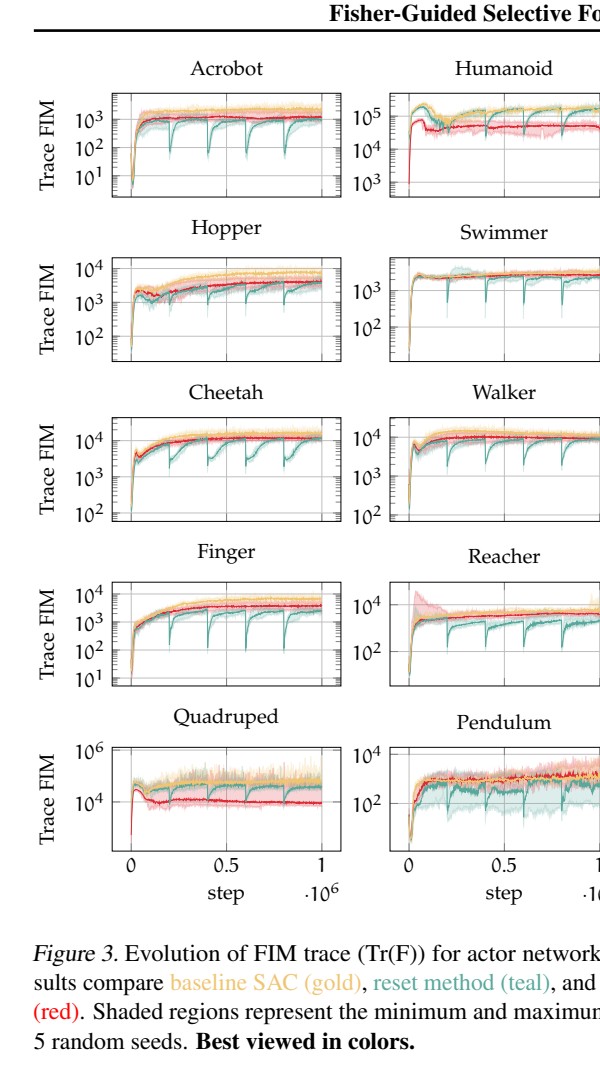

*Figure 3.* Evolution of FIM trace (Tr(F)) for actor networks. Results compare baseline SAC (gold), reset method (teal), and FGSF (red). Shaded regions represent the minimum and maximum over 5 random seeds. **Best viewed in colors.**

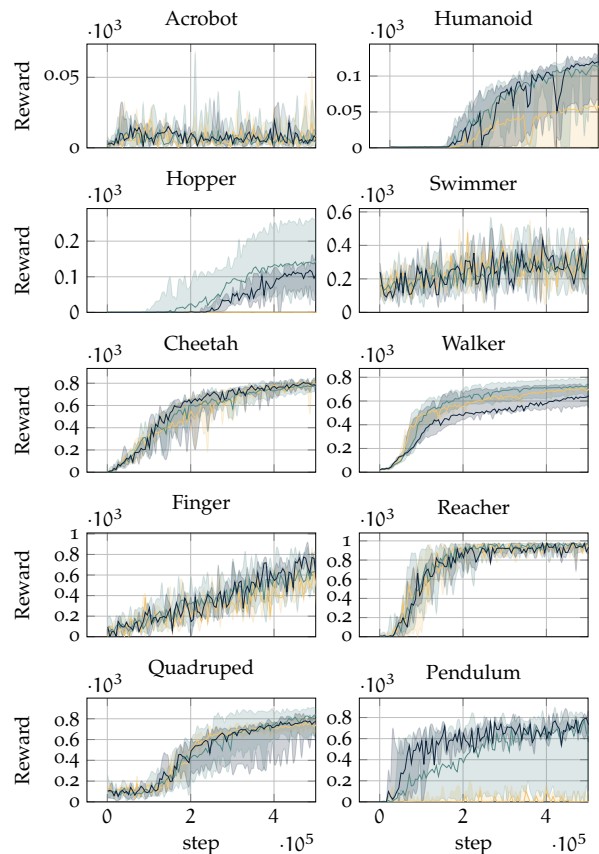

*Figure 4.* Hyperparameter sensitivity analysis showing performance across different scrubbing coefficients ($\lambda \in [5 \times 10^{-6}, 5 \times 10^{-8}]$). The lighter the color the higher the coefficient. Shaded regions represent the minimum and maximum over 5 random seeds. **Best viewed in colors.**

value to regularize the trace and further mitigate the effects of amplified early biases.

### 5.3. Ablation Studies

To dissect the contributions of different components to the FGSF method, we perform two ablation studies. First, we investigate the impact of network component scrubbing by selectively applying FGSF to either the critic-only, or both networks. Second, we analyze the influence of structured noise injection through a comparative evaluation against a simpler, unstructured approach.

**Impact of Network Component Scrubbing** Our investigation into critic-only scrubbing, shown in Figure 6, reveals that in complex, high-dimensional locomotion tasks like `Humanoid` and `Quadruped`, it achieves comparable, and sometimes better, performance than full network scrubbing, suggesting that the critic network is more susceptible to the PB. For example, in the `Humanoid` environment, critic-only scrubbing demonstrates more stable learning with fewer performance drops. In simpler envi-

ronments like `Pendulum` and `Reacher`, the difference between critic-only and full network scrubbing is minimal. However, in more complex environments like `Walker` and `Cheetah`, critic-only scrubbing shows improved stability in later stages of training,

FIM trace analysis in Figure 11 and 12 validates the superior effectiveness of critic-only scrubbing, showing more effective regularization during early training, with consistently lower Tr(F) values for both critic and actor networks. Notably, critic-only scrubbing achieves comparable, and in some cases superior, regularization of Tr(F) for the actor despite not directly manipulating its parameters, further emphasizing the critic's central role in PB development. Our analysis reveals an order-of-magnitude difference in Tr(F) values between critic and actor networks, revealing different operating regimes in parameter space, which is associated with the critic's role in value estimation. We refer the reader to Table 2 for a full panoramic of these results.

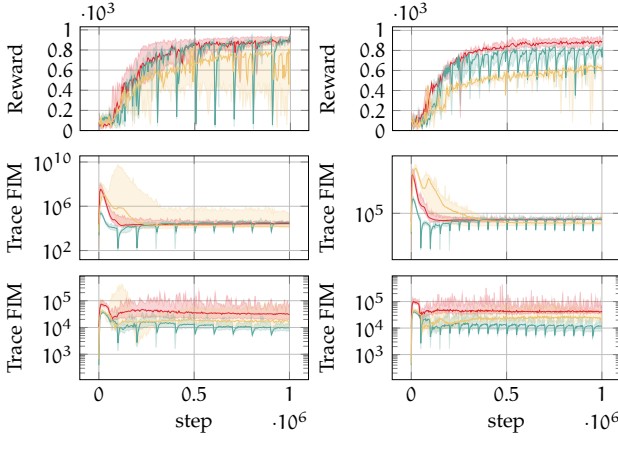

*Figure 5.* Performance comparison and FIM trace evolution on the `Quadruped` under different replay ratios (2 left column and 4 right column) for baseline SAC, reset method, and FGSF. Higher replay ratios amplify the differences between methods. **Best viewed in colors.**

**Fisher vs Gaussian Noise** To evaluate the importance of Fisher-guided noise injection, we conducted a comparative analysis between FGSF and a simpler Gaussian noise approach. The Gaussian noise variant samples perturbations from a distribution with a mean of 0 and a standard deviation equal to the mean of the network parameter values (i.e. $\mathcal{N}(0, 0.001\mu)$ where $\mu$ represents the mean of network parameter values). While multiple noise formulations were possible, this simple implementation provides a clear baseline. The results of this analysis are shown in Figure 7 In complex environments like `Humanoid` and `Quadruped`, FGSF showed modest performance improvements over the Gaussian Noise method while achieving significantly more stable learning trajectories. Although effective, Gaussian noise exhibits higher performance variance, especially in the `Humanoid` environment. This stability gap widens with increasing task dimensionality. In simpler environments like `Reacher` and `Pendulum`, both methods achieve similar final returns. However, FGSF maintains advantages in learning speed and stability. FGSF produces smoother learning curves than Gaussian noise injection, and learning dynamics show that FGSF achieves more consistent progress, suggesting more efficient parameter space exploration.

## 6. Discussion & Conclusion

This paper introduced Fisher-Guided Selective Forgetting, a novel method for mitigating the primacy bias in Deep Reinforcement Learning. By leveraging the Fisher Information Matrix and adapting techniques from machine unlearning, FGSF offers a principled approach to address the PB by selectively modifying network weights and controlling

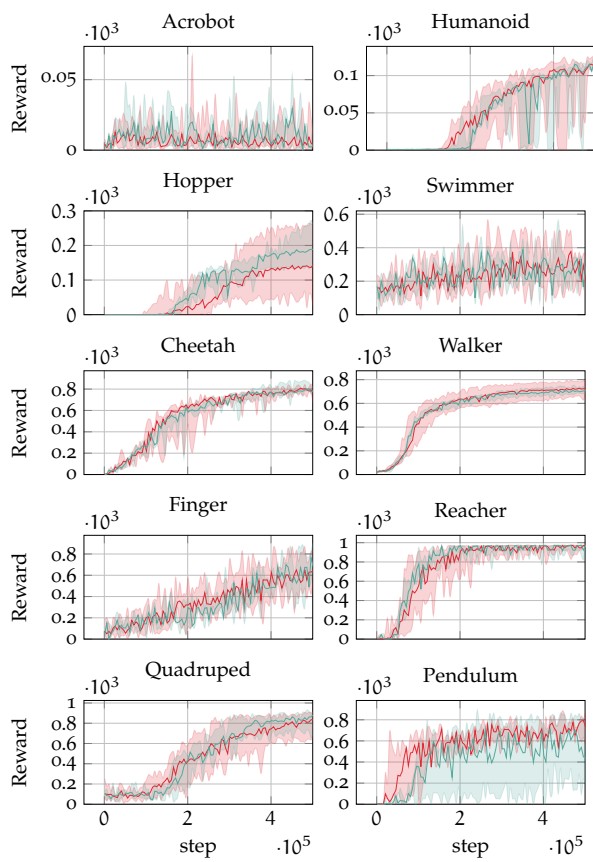

*Figure 6.* Learning curves showing episodic reward between critic-only scrubbing (red) and full network scrubbing (teal) for the different environments. **Best viewed in colors.**

the learning process. Our experiments, conducted across a diverse range of environments, demonstrate FGSF's effectiveness in several key areas. FGSF consistently achieves improved performance and stability compared to baseline SAC and the periodic network reset method, particularly in complex and high-dimensional tasks such as `Humanoid` and `Quadruped`, where we observed up to a $50\%$ increase in mean return compared to the baseline. Furthermore, our analysis highlights that the critic network is more susceptible to the PB than the actor, which aligns with previous studies (Lyle et al., 2022b;a; Van Hasselt et al., 2018), and that selectively addressing the critic's bias has a stronger impact on overall performance, allowing for more efficient computation. FGSF also demonstrated robustness across various replay ratios, maintaining performance stability even at a different replay ratio, where baseline SAC degraded significantly. We also showed an improvement compared to a simple Gaussian noise injection strategy. This might indicate that the geometric properties of the FIM can indeed be exploited for better-performing models and more effective bias mitigation.

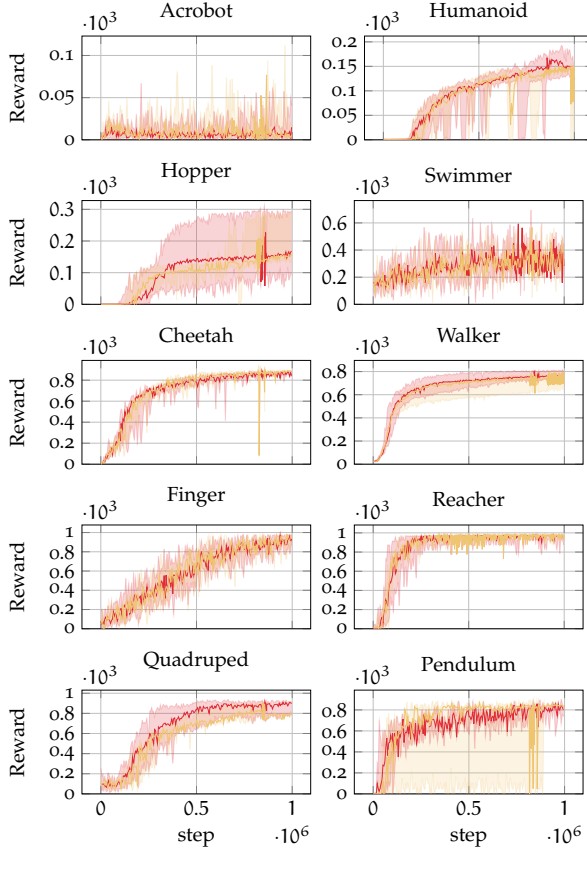

Figure 7. Performance comparison between FGSF and Gaussian noise injection across different environments. Shaded regions represent the minimum and maximum over 5 random seeds. **Best viewed in colors.**

Despite these encouraging results, it is essential to acknowledge the limitations of our approach. Firstly, while FGSF shows improved sample efficiency compared to baseline methods—achieving a $20\%$ reduction in the required samples to reach $90\%$ of the final performance in complex environments—the computation of the FIM still introduces a non-negligible overhead. Although we have used an efficient approximation of the FIM (EKFAC), the additional computational cost, which is between $10$-$20\%$ in cumulative training time (Figure 14), might be a practical concern for large-scale DRL applications or when computational resources are limited. Furthermore, while we have investigated the impact of the hyperparameter $\lambda$ and found optimal values around $5 \times 10^{-7}$, further research is needed for a more comprehensive analysis across a wider range of problems. Our observations also suggest a potential trade-off; simpler environments might not benefit as much from fine-tuning $\lambda$ and may even perform better with less regularization, indicating the need to adapt the scrubbing coefficient based on task complexity.

Our ablation study shows that even simple noise injection strategies, albeit not as effective as FGSF, can achieve significant performance improvements over the baseline SAC, indicating that the PB is indeed closely related to the optimization process itself. This resonates with the recent developments on continual backpropagation (Dohare et al., 2023), which suggest that directly manipulating the optimization process may be a promising approach to address similar problems in DRL. Furthermore, it suggests that future research might explore the effects of FGSF with alternative, potentially more sophisticated, optimization algorithms like natural gradient descent (Kakade, 2001; Pascanu, 2013), which is more closely aligned with the nature of the FIM.

Despite these limitations, our work opens up several interesting avenues for future research. The integration of machine unlearning techniques into the DRL framework represents a promising direction, creating a new family of algorithms that can selectively learn and unlearn from past experiences, potentially leading to more efficient and adaptable DRL agents. While our FGSF method demonstrates the value of structured information, further research could investigate alternative ways to leverage the FIM beyond simple noise injection, exploring different techniques of performing a weight update to achieve more targeted interventions. More work also needs to be done to better understand the interplay between the FIM trace, network plasticity, and capacity, particularly with regards to the critic's role. Finally, future work should explore more complex and diverse environments to better understand the limits of FGSF's applicability in more complex training scenarios. In this regard transfer learning comes to mind, where it has been shown that DRL agents often overfit on the source task they have been pre-trained on, and fail to adapt to the target task (Farebrother et al., 2018; Sabatelli & Geurts, 2021).

In conclusion, this paper contributes a novel approach, FGSF, for addressing the primacy bias in DRL by exploiting the theoretical framework of information geometry and machine unlearning. Our findings demonstrate the potential of integrating FIM-based techniques for a better understanding and mitigation of biases in neural networks and open new directions for research and future work, in the continuous quest for better and more robust DRL systems

## 7. Impact Statement

This paper presents work whose goal is to advance the field of Machine Learning. There are many potential societal consequences of our work, none which we feel must be specifically highlighted here.

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

## A. Supplementary Material

The supplementary material contains the code necessary to reproduce all experiments and analyses presented in this work. This includes scripts for data preprocessing, and model training allowing readers to independently verify our findings.

## B. Comparative Analysis of FGSF

*Table 1.* Performance comparison of different algorithms (FGSF, Reset method, and Baseline SAC) across various environments. Values represent the mean and standard deviation of the final 100 episode returns over 5 random seeds. Magenta represents the best performing algorithm.

| Environment | FGSF | Reset | Base SAC |
|---|---|---|---|
| Acrobot | 6.481 ± 2.823 | 4.056 ± 2.449 | 145.313 ± 24.776 |
| Humanoid | 136.645 ± 14.360 | 91.539 ± 31.977 | 68.503 ± 21.938 |
| Hopper | 148.024 ± 7.215 | 149.689 ± 47.105 | 266.906 ± 18.120 |
| Swimmer | 326.493 ± 65.572 | 284.452 ± 59.983 | 324.661 ± 56.188 |
| Cheetah | 838.635 ± 24.309 | 541.672 ± 249.813 | 803.851 ± 65.473 |
| Walker | 746.695 ± 13.214 | 599.996 ± 148.548 | 758.397 ± 20.905 |
| Finger | 824.243 ± 77.102 | 279.302 ± 103.450 | 855.262 ± 66.543 |
| Reacher | 958.788 ± 17.978 | 899.837 ± 205.015 | 940.631 ± 33.600 |
| Quadruped | 873.473 ± 21.287 | 688.864 ± 152.444 | 582.909 ± 37.262 |
| Pendulum | 770.519 ± 51.891 | 513.362 ± 149.628 | 834.720 ± 12.865 |

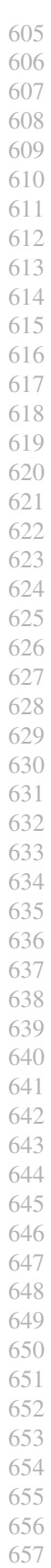

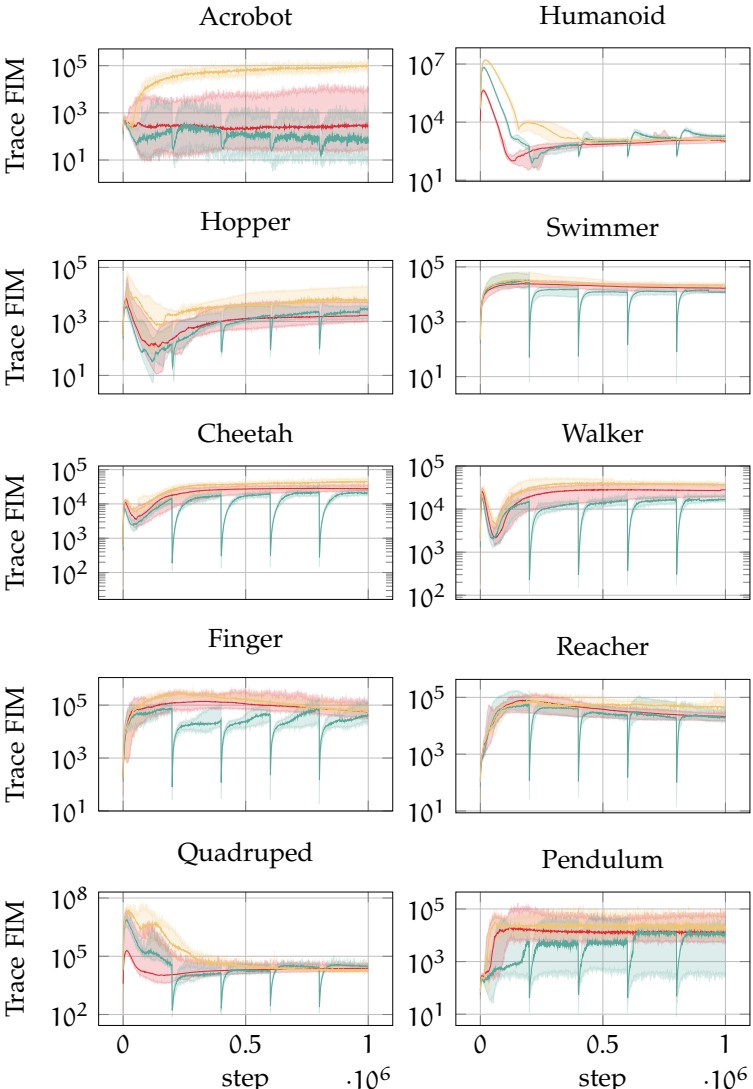

*Figure 8.* Evolution of FIM trace (Tr(F)) during training for critic networks across different environments. Results compare baseline SAC (gold), reset method (teal), and FGSF (red). Shaded regions represent the minimum and maximum over 5 random seeds. **Best viewed in colors.**

## B.1. Weight Update Magnitude

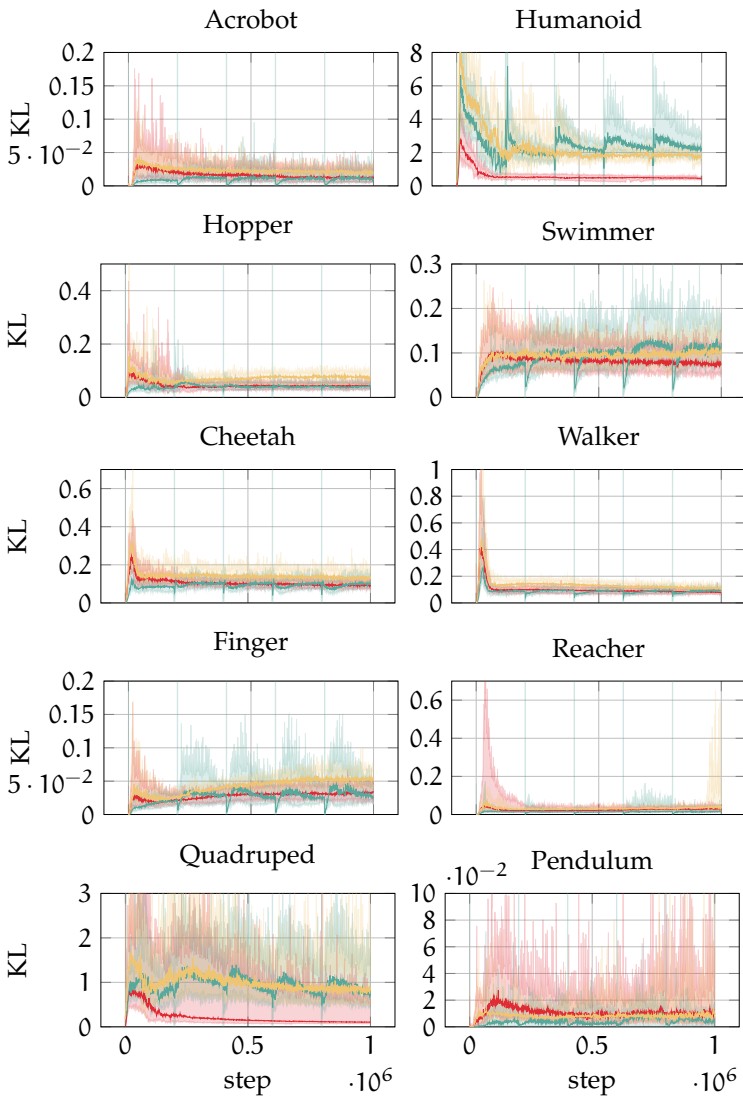

*Figure 9.* Local parameter update magnitudes measured by KL divergence across different environments. Lower values indicate a smaller parameter update. Spikes in the baseline (gold) and reset methods (teal) contrast with FGSF's (red) more consistent update pattern. **Best viewed in colors.**

Our analysis of parameter update magnitudes, measured by the Kullback-Leibler (KL) divergence of weight distributions, reveals that in complex environments, FGSF maintains consistently lower update magnitudes (local delta) throughout training (typically stabilizing between 0.5 and 0.7), with smoother trajectories compared to the higher values and more pronounced spikes observed in baseline SAC. While `Cheetah` and `Swimmer` show periodic spikes, FGSF maintains better stability. These results suggest that FGSF's improved performance is partly due to controlled parameter updates, preventing destabilizing policy changes.

## B.2. Dormant neurons

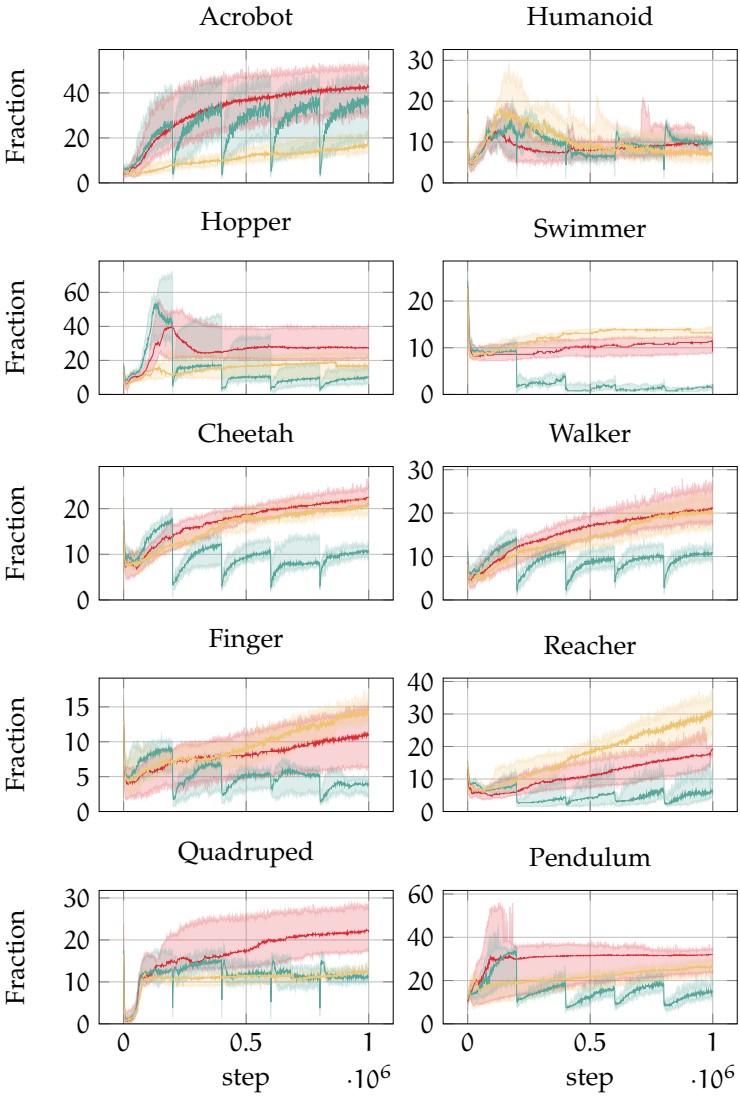

*Figure 10.* Fraction of dormant neurons during training across different environments. Plots compare baseline SAC (gold), reset method (teal), and FGSF (red). **Best viewed in colors.**

In baseline SAC, critic networks exhibit a consistent increase in dormant neuron fraction, particularly in complex environments. In the `Quadruped` environment, this rises from 2% to approximately 6% while in the `Humanoid` environment, it reaches peaks of 8% before stabilizing around 4%. This progressive loss of active neurons correlates strongly with Tr(F) stabilization, suggesting a link between the identified learning phases and network plasticity. FGSF, despite achieving superior performance, either matches or exceeds the baseline in terms of dormant neuron fraction, challenging the idea that dormant neuron fraction is a reliable indicator of the primacy bias.

# C. Impact of Network Component Scrubbing

*Table 2.* Comparison of learning curves between critic-only scrubbing and full network scrubbing for the different environments. Values represent the mean and standard deviation of the final 100 episode returns over 5 random seeds. Magenta represents the best performing algorithm

| Environment | Critic-only Scrubbing | Full Scrubbing |
| --- | --- | --- |
| Acrobot | 6.534 ± 3.019 | 15.105 ± 12.254 |
| Humanoid | 137.800 ± 13.899 | 134.148 ± 9.096 |
| Hopper | 150.433 ± 22.456 | 229.593 ± 34.675 |
| Swimmer | 333.186 ± 74.840 | 324.078 ± 78.075 |
| Cheetah | 842.421 ± 22.509 | 828.527 ± 27.339 |
| Walker | 749.690 ± 13.521 | 746.717 ± 28.791 |
| Finger | 833.518 ± 73.105 | 769.001 ± 104.140 |
| Reacher | 959.554 ± 17.695 | 962.087 ± 20.241 |
| Quadruped | 873.464 ± 21.803 | 889.924 ± 24.096 |
| Pendulum | 771.567 ± 53.413 | 660.094 ± 108.702 |

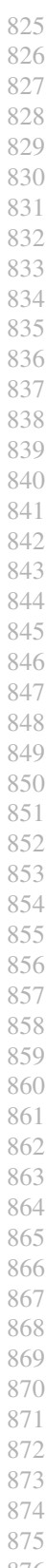
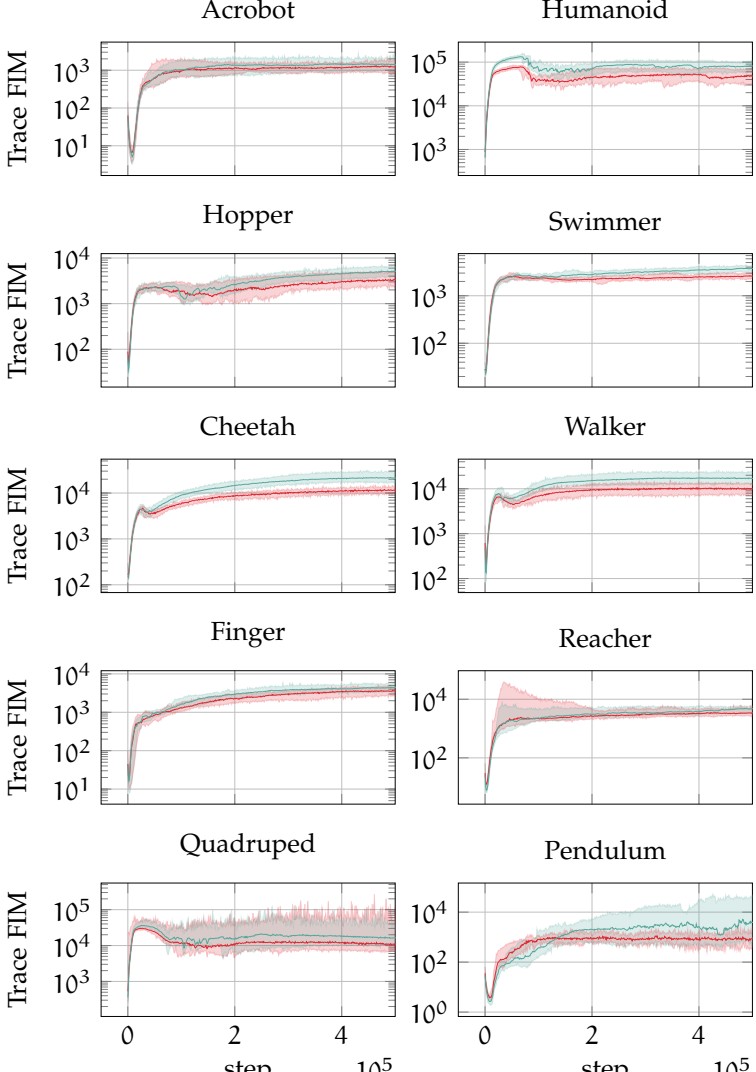

*Figure 11.* Actor network FIM trace evolution comparing critic-only scrubbing (red) versus full network scrubbing (teal) for different environments. Results demonstrate that critic-only scrubbing achieves effective regularization of actor network dynamics even without direct intervention. **Best viewed in colors.**

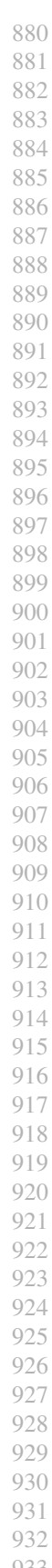
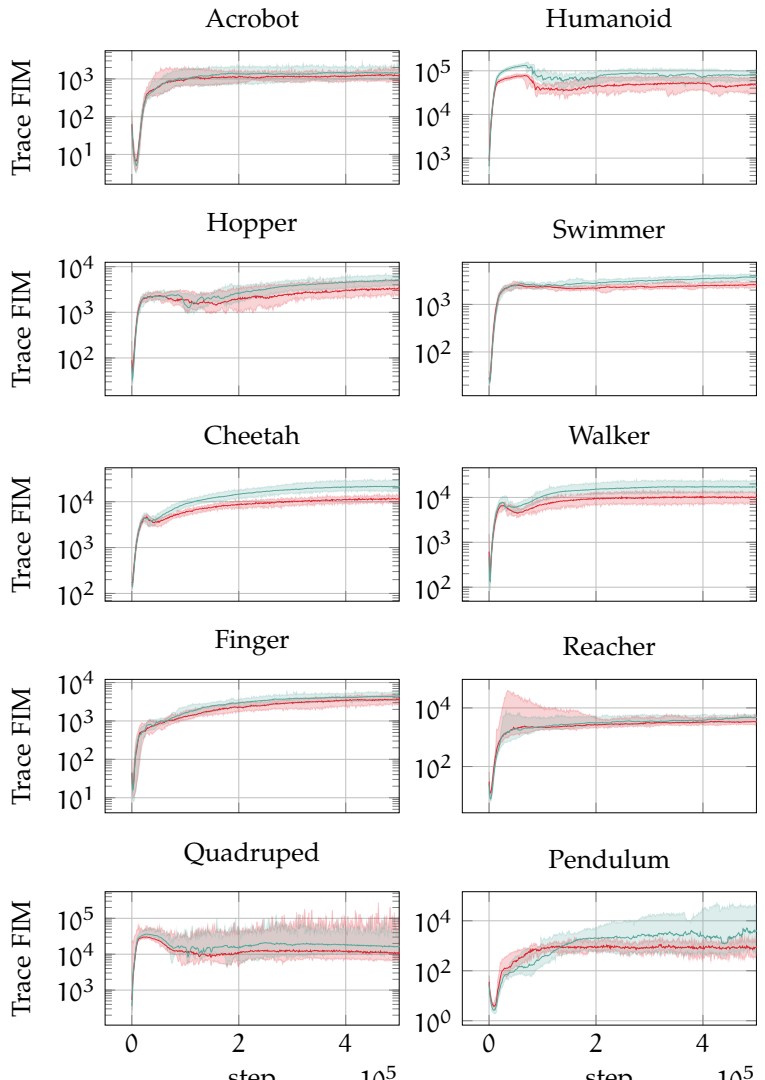

*Figure 12.* Critic network FIM trace evolution under critic-only scrubbing (red) versus full network scrubbing (teal) for different environments. The traces show stronger regularization effects in critic-only scrubbing. **Best viewed in colors.**

# D. Hyperparameter Sensitivity

*Table 3.* Hyperparameter sensitivity analysis showing performance across different scrubbing coefficients ($\lambda$). Values represent the mean and standard deviation of the final 100 episode returns over 5 random seeds. Magenta represents the best performing algorithm

| Environment | $\lambda = 5 \times 10^{-6}$ | $\lambda = 5 \times 10^{-7}$ | $\lambda = 5 \times 10^{-8}$ |
|---|---|---|---|
| Acrobot | 4.915 ± 3.419 | 6.581 ± 3.033 | 7.728 ± 4.624 |
| Humanoid | 1.246 ± 0.100 | 137.800 ± 13.899 | 129.153 ± 17.056 |
| Hopper | 0.026 ± 0.058 | 148.061 ± 19.055 | 132.705 ± 16.831 |
| Swimmer | 316.372 ± 68.958 | 331.688 ± 74.640 | 320.555 ± 63.527 |
| Cheetah | 851.145 ± 16.972 | 826.530 ± 22.699 | 854.613 ± 16.708 |
| Walker | 729.304 ± 16.350 | 749.204 ± 13.887 | 715.562 ± 27.385 |
| Finger | 818.001 ± 106.650 | 835.705 ± 72.117 | 872.998 ± 73.859 |
| Reacher | 957.382 ± 22.751 | 959.448 ± 17.840 | 968.659 ± 18.319 |
| Quadruped | 775.383 ± 20.572 | 874.090 ± 21.353 | 861.912 ± 28.982 |
| Pendulum | 55.007 ± 58.313 | 725.280 ± 66.427 | 770.626 ± 54.307 |

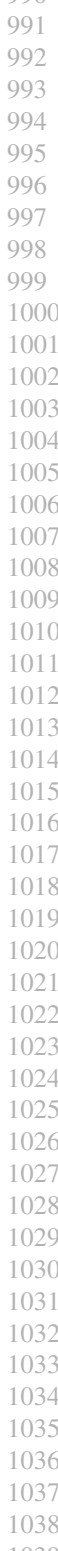
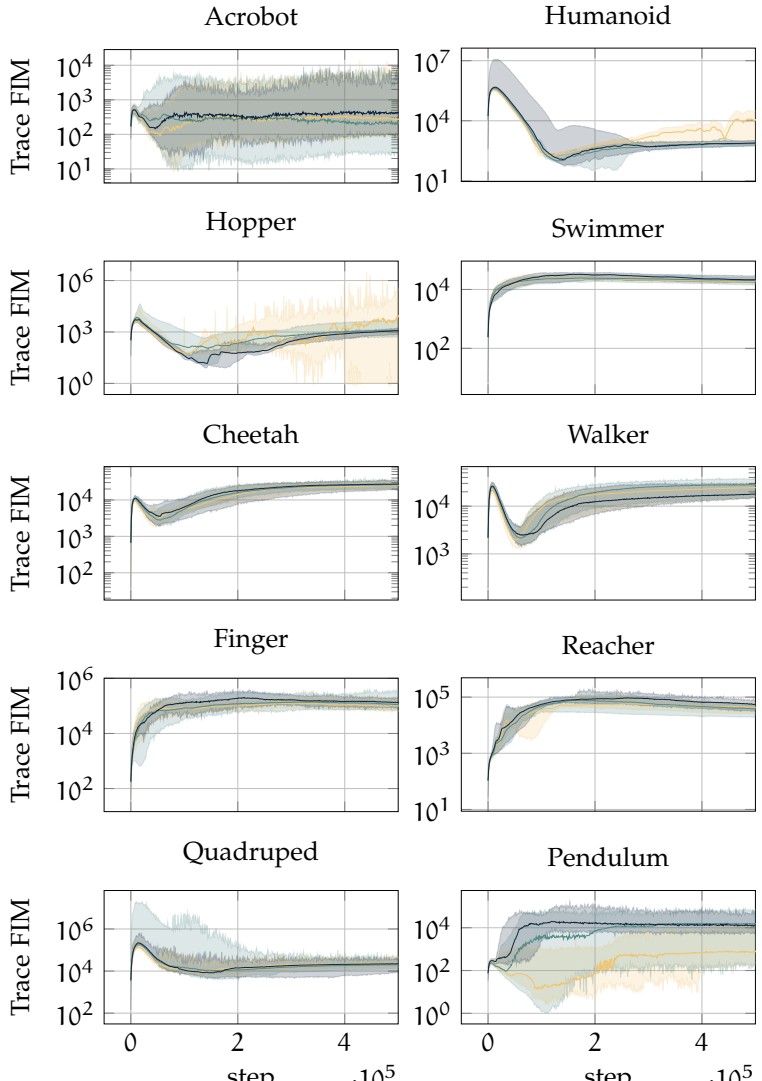

*Figure 13.* FIM trace of the critic network under different scrubbing coefficients ($\lambda \in [5 \times 10^{-6}, 5 \times 10^{-8}]$), illustrating the relationship between $\lambda$ values and the FIM trace. The lighter the color the higher the coefficient. **Best viewed in colors.**

# E. Computational Considerations

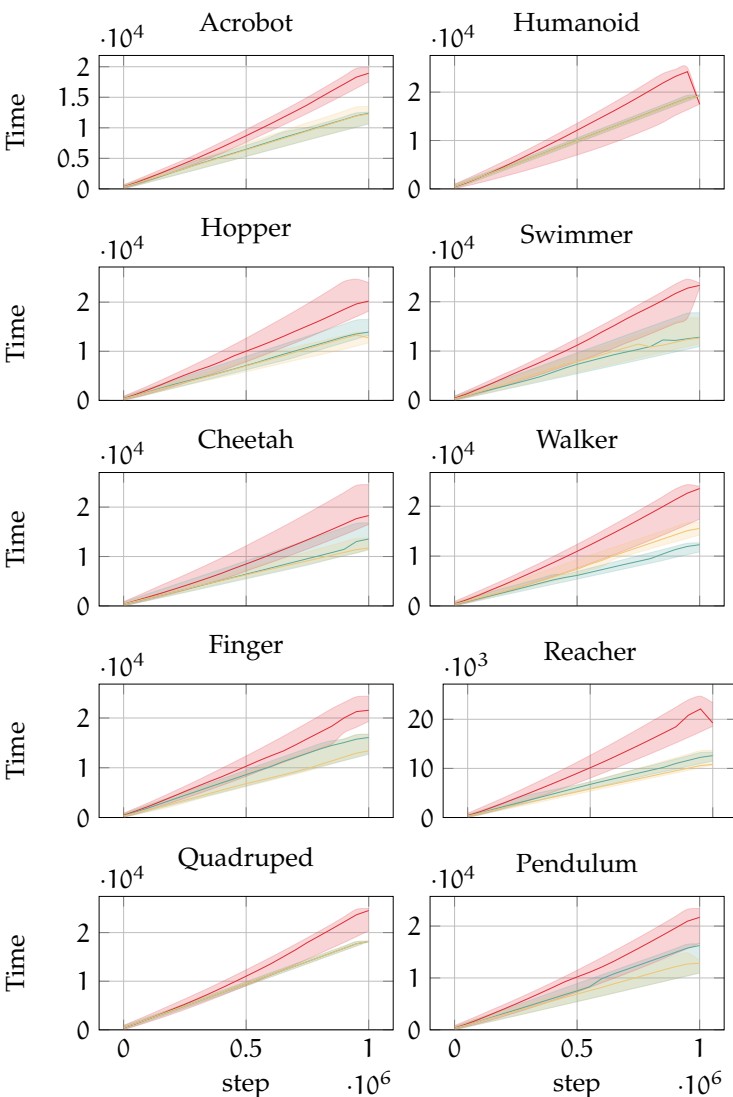

*Figure 14.* Comparative analysis of cumulative training time across environments. The y-axis shows total computation time in seconds, demonstrating the computational overhead of different methods. Baseline SAC (gold), reset (teal) and FGSF (red). **Best viewed in colors.**

FGSF shows a 15-20% increase in cumulative update time compared to baseline SAC in high-dimensional environments like `Humanoid` and `Quadruped`. This overhead remains relatively constant throughout training, as evidenced by parallel slopes in the timing curves. The reset method has no computational overhead.

