# OpenReview forum: "Fisher-Guided Selective Forgetting: Mitigating Primacy Bias in Deep Reinforcement Learning"
_ICML.cc/2025/Conference — Submitted to ICML 2025_

### Official Review · Reviewer_738E · 2025-03-12

**Overall Recommendation:** 1

**Summary:**

This paper proposes to address primacy bias (early experiences having an adverse effect on later training) in reinforcement learning by using a forgetting mechanism. The proposed method utilizes an update rule from the _machine unlearning_ literature to forget past experiences and reduce primacy bias. The updates utilize the empirical Fisher to scale Gaussian noise to be added to the weights. This method is evaluated on standard continuous control benchmark tasks.

**Claims And Evidence:**

- The proposed method seems to indeed outperform the base SAC agent on the benchmark tasks, but it is unclear whether it is better than adding simple Gaussian noise without rescaling by the empirical Fisher. In Fig.7, the shaded regions largely overlap so there is no clear difference in performance. This casts doubt on the importance of using the empirical Fisher.

- Another concern is related to how resets do not seem to provide a benefit to the base SAC agent. See "experimental designs" section for more details.

**Essential References Not Discussed:**

The paper "Directions of Curvature as an Explanation for Loss of Plasticity" Lewandowski et al. (2024) proposes a hypothesis closely related to the idea of the evolution of the Fisher information indicating primacy bias. In that paper, they discuss how a loss of curvature (as measured by the Hessian rank) can be linked to loss of plasticity (which is closely linked to primacy bias). In fact, they also check the spectrum of a matrix of gradient outer products (which is similar to the definition of the empirical Fisher). There is a direct relationship between the spectrum and the trace since the trace is the sum of the eigenvalues.

**Experimental Designs Or Analyses:**

The benchmark tasks and baseline may need to be chosen more carefully. In many of the tasks presented, resets are not providing any benefit over the baseline SAC agent. This undermines the main idea that there is primacy bias that needs to be addressed through the proposed method.

There is a section on using higher replay ratios where there does seem to be more of a benefit to resets (suggesting primacy bias is present). It may be better to focus on this setting. In [1], we can see a larger benefit to resets with larger replay ratios.

Alternatively, reconsidering which tasks to run and choosing those where resets provide some benefit could be an alternative approach.

[1] "Sample-Efficient Reinforcement Learning by Breaking the Replay Ratio Barrier" D'Oro et al. (2023)

**Methods And Evaluation Criteria:**

- Using the Deepmind Control Suite with SAC is a suitable benchmark. Adding resets to SAC is a reasonable baseline.

- Other baselines from the plasticity loss literature should be tested.  For example, Shrink-and-perturb [1], l2-init [2], Parseval regularization [3], Spectral regularization [4] to name a few choices. These methods all update weights to improve plasticity and reduce primacy bias (or add a regularization term to accomplish this) and would help assess the relative utility of the proposed method.

[1] "On Warm-Starting Neural Network Training" Ash and Adams (2019) \
[2] "Maintaining Plasticity in Continual Learning via Regenerative Regularization" Kumar et al. (2023) \
[3] "Parseval Regularization for Continual Reinforcement Learning" Chung et al. (2024) \
[4] "Learning continually by spectral regularization" Lewandowski et al. (2024)

**Other Comments Or Suggestions:**

- An analysis of the empirical Fisher could be interesting. In particular, looking at which dimensions it emphasizes would be interesting to identify which weights are considered to be forgottten or remembered. Comparing this to other common heuristics for weight importance such as weight magnitude (related to the neuron importance measure in [1]).

- Fig. 4 should have a legend to indicate the hyperparmeter values for each curve.

Minor questions:
- What is $\sigma^2$ in the update? Is this estimated or is it combined with $\lambda$ into a single hyperparameter. Clarify this in the paper.

- Why plot max/min in shaded regions instead of confidence intervals?


[1] "Continual Backprop: Stochastic Gradient Descent with Persistent Randomness" Dohare et al.

**Other Strengths And Weaknesses:**

- The idea of using methods from the machine unlearning literature is interesting and novel. Perhaps there could be more connections drawn there or alternative methods from that space could be tried instead.

- Generally, the writing is clear and the presentation is nice.

- The paper would be benefit from more exposition to the area of machine unlearning. Since this paper is written for a RL audience, a reader is likely to be unaware of these works. Currently, section 3.3 requires additional explanations. Explaining the loss defined there and some of the terms such as $P(S(w)|D)$ would be necessary.

**Questions For Authors:**

None

**Relation To Broader Scientific Literature:**

This paper highlights an interesting opportunity to use ideas from the _machine unlearning_ literature to address primacy bias. The evaluation is currently not thorough enough to properly assess the proposed method, although there is some promise.

**Theoretical Claims:**

N/A

---

> ### Author Rebuttal · Authors · 2025-03-31
>
> We thank Reviewer 738E for their detailed feedback. We address their main concerns below.
>
> ### FGSF vs Gaussian Noise:
>
> We appreciate the reviewer's analysis of the FGSF vs Gaussian noise comparison (Fig 7).
>
> **Ablation Goal:** This ablation showed general noise injection strategies offer benefits over baseline SAC, suggesting noise can combat plasticity loss/Primacy Bias (PB).
>
> **FGSF's Advantage:** While the average performance difference across environments seems marginal (Fig 7), FGSF's key benefit is its targeted, FIM-guided noise. This advantage is clearer in:
>
> * **High-PB Environments:** In tasks like Humanoid/Quadruped with strong PB (Fig 3/8), FGSF offers more consistent stability and often better peak performance than Gaussian noise (compare Fig 7 vs Fig 2).
> * **Importance of FIM:** The FIM allows strategic noise injection (adapting to network sensitivity/geometry), unlike simple Gaussian noise. This geometric insight is valuable.
> ### Experimental Design:
>
> We acknowledge the reset baseline's benefit over SAC isn't consistently strong at RR=1.
>
> **PB Strength Varies:** PB strength depends on task complexity and settings like RR. In simpler tasks or at RR=1, PB may be less pronounced, reducing the reset's observed benefit.
>
> **High RR Confirmation:** As the reviewer and D'Oro et al. (2023) note, PB is often exacerbated at higher RRs. Our RR=4 results (Fig 5) confirm this: baseline SAC degrades, the reset method shows some unstable benefit, and FGSF provides significant, stable improvements over both. This shows FGSF effectively addresses amplified PB.
>
> **Focus:** Focusing experiments on settings with strong PB (complex tasks or high RR) better highlights the problem FGSF solves. The RR=1 results were included as the standard from previous work (e.g., Haarnoja et al., 2018).
> ### Methods:
>
> We thank the reviewer for suggesting baselines (Shrink-and-perturb, l2-init, Parseval/Spectral reg). We acknowledge these related works. Comparing FGSF to these requires substantial effort. Our initial goal was introducing the FIM-geometric perspective and establishing viability against the reset baseline.
>
> ### Weaknesses:
> We agree more background on machine unlearning (Sec 3.3) is needed for the RL audience. We will expand this section in the camera-ready version to clarify the Forgetting Lagrangian ($L$), distributions $P(\cdot|\cdot)$, the KL term, and λ.
> ### Response to Minor Questions:
> **FIM Dimension Analysis:** Analyzing which dimensions the FIM emphasizes is interesting future work but beyond the current scope.
>
> **Fig 4 Legend:** We will add a legend indicating λ values to Figure 4.
>
> **$\sigma^2$ in Update:** $\sigma^2$ comes from the uncertainty term in Golatkar et al.'s derivation. In our FGSF algorithm (Eq. Sec 3.4), we tune λ. The $(\sigma^2)^{1/4}$ factor is absorbed into our effective λ. We will clarify this.
>
> **Min/Max Shading:** We used min/max shading (5 seeds) to show the full performance variability (worst/best cases). With few seeds (common in DRL), standard CIs can be unreliable, and performance distributions may not be normal. Min/max shading avoids distributional assumptions and shows the empirical boundaries found, offering valuable information in this low-sample context.
>
> We thank the reviewer again for their assessment. We hope these responses clarify FGSF's contributions, particularly where PB is pronounced. We will incorporate several improvements based on this feedback. We hope our clarifications provide grounds for reconsideration despite the Reject recommendation. Given space constraints, we welcome follow-up questions to address any remaining concerns in more detail.
>
> ### References:
> 1. Haarnoja, Tuomas, Aurick Zhou, Pieter Abbeel, and Sergey Levine (2018). “Soft actor-critic: Off-policy maximum entropy deep reinforcement learning with a stochastic actor.”
> 2. Golatkar, Aditya, Alessandro Achille, and Stefano Soatto (2020a). “Eternal sunshine of the spotless net: Selective forgetting in deep networks.”
> 3. D’Oro, Pierluca, Max Schwarzer, Evgenii Nikishin, Pierre-Luc Bacon, Marc G Bellemare, and Aaron Courville (2022). “Sample-efficient reinforcement learning by breaking the replay ratio barrier.” In: Deep Reinforcement Learning Workshop NeurIPS 2022.
> 4. Dohare, Shibhansh, J Fernando Hernandez-Garcia, Parash Rahman, ARupam Mahmood, and Richard S Sutton (2023). "Maintaining plasticity in deep continual learning.”
> 5. Hochreiter, Sepp and Jürgen Schmidhuber (1997). “Flat minima.”
> 6. Jastrzebski, Stanislaw, Devansh Arpit, Oliver Astrand, Giancarlo BKerg, Huan Wang, Caiming Xiong, Richard Socher, Kyunghyun Cho, and Krzysztof J Geras (2021). “Catastrophic fisher explosion:Early phase fisher matrix impacts generalization.”
> 7. Obando-Ceron, Johan, Aaron Courville, and Pablo Samuel Castro (2024). “In value-based deep reinforcement learning, a pruned network is a good network.”
> 8. Martens, James and Roger Grosse (2015). “Optimizing neural networks with kronecker-factored approximate curvature.”

---

### Official Review · Reviewer_LhZa · 2025-03-13

**Overall Recommendation:** 4

**Summary:**

Deep reinforcement learning (RL) agents tend to overfit their early experiences, limiting their ability to learn from subsequent interactions. Nikishin et al. (2022) identified this phenomenon as primacy bias. While they highlighted the issue, their proposed solution—periodically resetting part of the network—is simple and lacks a mechanism to characterize primacy bias. To address these limitations, the authors propose using the trace of the Fisher Information Matrix (FIM) to characterize primacy bias. They also introduce Fisher-Guided Selective Forgetting (FGSF), a method to selectively update the weights to mitigate primacy bias. They show the efficacy of their method empirically on DM Control Suite and add some ablation studies investigating the effectiveness of their introduced components.  They demonstrate the efficacy of their method empirically on the DM Control Suite and conduct ablation studies to evaluate the effectiveness of its components.

**References**
- Nikishin, E., Schwarzer, M., D’Oro, P., Bacon, P. L., & Courville, A. (2022, June). The primacy bias in deep reinforcement learning. In International conference on machine learning (pp. 16828-16847). PMLR.

**Claims And Evidence:**

There are claims in the paper that are lacking in evidence and/or not clear to me.

> Claim 1: The Trace of the Fisher Information Matrix (FIM) characterizes primacy bias.

I do not see how the authors make this claim from Fig.1. If anything, it seems that the trace of the FIM is unrelated to performance.
While the the trace of the FIM does give a sense of overall sensitivity of the network’s parameters, there is no evidence in the learning curves indicating primacy bias! Quadruped doesn't seem to suffer from primacy bias. Is this observation correct?

> Claim 2: Fisher-Guided Selective Forgetting (FGSF) is a principled mitigation strategy that relies on a geometric understanding of the primacy bias to selectively modify network weights

I do not understand what the authors mean by principled here. In sub-section 3.4, they mention that the optimal forgetting procedure takes the form: $S(w) = w − B^{−1} \nabla L_{Dr} (w) + (\lambda \sigma^2)^{1/4} B^{−1/4} \epsilon$,
where $B$ is the Hessian of the loss on the retained data, $\epsilon$ is standard Gaussian noise, and $\sigma^2$ represents the uncertainty.

In sub-Section 3.5, they propose their approach: $S(w) = w +  (\lambda \sigma^2)^{1/4} F^{−1/4} \epsilon$, where FIM is the empirically estimated FIM.

Why is dropping a term and replacing the Hessian with Fisher Information Matrix a good approximation? Can you elaborate further? Also, why is this principled?

**References**
- Golatkar, A., Achille, A., and Soatto, S. Eternal sunshine of the spotless net: Selective forgetting in deep networks. In
Proceedings of the IEEE/CVF Conference on Computer Vision and Pattern Recognition, pp. 9304–9312, 2020a.

**Essential References Not Discussed:**

N/A

**Experimental Designs Or Analyses:**

- The plots intended to motivate the use of the trace of the FIM for characterizing primacy bias are unconvincing. Figure 1 does not provide strong support, and Figure 3 is even more problematic, as this metric suggests that the proposed method, FGSF, performs worse than the baseline SAC on average in terms of Tr(FIM)

- In Fig. 9, the authors claim that "... FGSF maintains consistently lower update magnitudes (local delta) throughout training (typically stabilizing between 0.5 and 0.7), with smoother trajectories compared to the higher values and more pronounced spikes observed in baseline SAC". This is not true! It's true on a couple of environments and false on others

My main concern is that it's unclear why I should consider using FGSF. It doesn't seem to help maintain a healthy Tr(F) or improve performance. Can the authors clarify why this method is important?

**Methods And Evaluation Criteria:**

I have some concerns about the choice of competitors used for comparison against FGSF:

1. Have the authors considered methods from recent studies on the loss of plasticity in deep learning? How would these methods perform in this setting? Specifically, have you considered approaches such as (1) Continual Backprop (Dohare et al., 2024), (2) Layer Norm + Weight Decay (Lyle et al., 2024), and (3) ReDo (Sokar et al., 2023)?

2. While the authors discuss primacy bias, their environments do not exhibit a strong "priming" effect. I suggest adding 1-2 additional environments with inherent non-stationarity to better demonstrate the advantages of their approach. My intuition is that FGSF would outperform its competitors in such settings.


**References**
- Dohare, S., Hernandez-Garcia, J. F., Lan, Q., Rahman, P., Mahmood, A. R., & Sutton, R. S. (2024). Loss of plasticity in deep continual learning. Nature, 632(8026), 768–774.
- Lyle, C., Zheng, Z., Khetarpal, K., Martens, J., van Hasselt, H. P., Pascanu, R., & Dabney, W. (2024). Normalization and effective learning rates in reinforcement learning. Advances in Neural Information Processing Systems, 37, 106440-106473.
- Sokar, G., Agarwal, R., Castro, P. S., & Evci, U. (2023, July). The dormant neuron phenomenon in deep reinforcement learning. In International Conference on Machine Learning (pp. 32145-32168). PMLR.

**Other Comments Or Suggestions:**

If the authors provide additional empirical evidence—such as comparisons with relevant competitors on 2/3 environments—during the rebuttal and address my other concerns, I would be willing to raise my score.

Typos:
- In Line 121, the references are missing the year
- The first line of subsection 3.1 is incomplete - "The FIM is a fundamental concept in information geometry that quantifies the amount of information a random variable carries about an unknown parameter". There is not mention of the relationship between the parameter (e.g., $\theta$) upon which the probability of the random variand $\displaystyle X$ depends.
- Line 228 (left) is a repetition of line 211 (right)

**Other Strengths And Weaknesses:**

I believe this work addresses an important issue in deep RL, and I encourage the authors to invest more effort in their writing and experiment selection to better convey their ideas. This has the potential to be a very interesting line of research. I also like their idea to leverage FIM as a tool to characterize the local geometry of the parameter space and measure network sensitivity.

**Questions For Authors:**

1. Have the authors considered methods proposed in recent papers on the loss of plasticity in deep learning (mentioned in Methods And Evaluation Criteria)? How would those methods perform in this context?
2. Is primacy bias a manifestation of the loss of plasticity?
3. How do the authors propose extending their method to on-policy methods, such as PPO?
4. Are large replay buffers necessary for FGSF to be effective? Is FGSF reliant on forgetting and re-learning frequently enough to achieve good performance?
5. The authors mention using a Savitzky-Golay filter in Subsection 3.2 to calculate the trace of the FIM. Could they elaborate on why this specific method was chosen?
6. I need some clarification regarding the algorithm pseudocode:
   a) In lines 7-9, is the agent scrubbing the data sampled from the replay buffer?
   b) What does N represent— the size of the replay buffer or the size of the mini-batch?
7. $\lambda \sim [5 \times 10^{-6}, 5 \times 10^{-8}]$: These values are extremely small. Is it due to the magnitude of the FIM? If so, normalizing the observations, features, and rewards could help (Vasan et al., 2024).
8. In Figure 10, why does FGSF have so many dormant neurons? How is this not affecting performance?

**References**
- Vasan, G., Elsayed, M., Azimi, S. A., He, J., Shahriar, F., Bellinger, C., ... & Mahmood, R. (2024). Deep policy gradient methods without batch updates, target networks, or replay buffers. Advances in Neural Information Processing Systems, 37, 845-891.

**Relation To Broader Scientific Literature:**

This paper fits with the broader class of papers on primacy bias in deep RL and loss of plasticity in continual learning literature.

**Theoretical Claims:**

N/A

---

> ### Author Rebuttal · Authors · 2025-03-31
>
> We thank Reviewer LhZa for their constructive criticism.
>
> **Claim 1**
>
> Following from our response to Reviewer PPSq regarding FIM interpretation (Weakness 3). Figure 1 primarily illustrates the two-phase pattern described in Section 3.2. The core evidence linking Tr(F) dynamics to PB and performance comes from:
> * **The Two-Phase PB Signature**: As detailed in Sec 3.2, PB manifests as an initial sharp increase in Tr(F), indicating rapid overfitting to early data, followed by a decrease.
> * **Correlation with Performance**: Figures 3 & 8 show that in environments like Quadruped where baseline SAC performs poorly due to PB, it exhibits an early Tr(F) spike. FGSF prevents this, maintaining a lower, controlled trace, and achieves significantly better performance. This aligns with findings linking high early Tr(F) to poor generalization and lower Tr(F) to better generalization via flatter minima (Jastrzebski et al., 2021).From these two points, it follows that Quadruped does suffer from PB. Therefore, the claim is not that Tr(F) directly equals performance, but that its dynamic pattern, particularly the problematic early spike, is a quantifiable signature of the PB phenomenon that correlates strongly with subsequent performance limitations.
>
> **Claim 2**
> * **Hessian → FIM**: Replacing the Hessian (B) with the empirical FIM is a standard and computationally feasible approximation in deep learning (Martens & Grosse, 2015). The FIM captures crucial second-order information related to the local geometry.
> * **Dropping $\nabla L_{Dr}$ Term**: This term from the original formulation is omitted in our adaptation for two key reasons specific to the periodic RL application : 1) The standard RL gradient update step already performs a similar parameter update based on the current batch loss. 2) Unlike one-shot unlearning, FGSF is applied periodically during training, making this specific term redundant. This adaptation makes the principled approach tractable and effective in DRL
>
> **Environments**
> Would tasks like the bit-flipping problem (Dohare et al., 2024) be representative of what the reviewer has in mind?
>
> **Tr(FIM) Plots (Fig 1/3):**
> As clarified under Claim 1, the key insight from the FIM trace is not that "lower is always better globally", but that the pathological early spike in baseline methods (Fig 3/8) signifies PB-induced overfitting.
>
> **KL Divergence (Fig 9):**
> "... reveals that in complex environments, FGSF maintains consistently lower ..." This is true in complex environments (e.g. Humanoid)
>
> **Why Use FGSF?:**
> FGSF offers an approach to PB mitigation grounded in information geometry, moving beyond heuristic resets. Its importance lies in: (1) Providing a framework to characterize PB via FIM dynamics. (2) Demonstrating significant stability improvements over reset methods. (3) Achieving superior final performance in complex, PB-prone environments (Table 1, Fig 2). (4) Offering better sample efficiency than baseline SAC (Sec 5.1).
>
> **Methods for Plasticity Loss:** We acknowledge these methods and are planning comparisons.
>
> **PB vs Plasticity Loss:** They are deeply related. PB, the overfitting to early data, is arguably a primary cause or specific manifestation of the broader phenomenon of plasticity loss (the network becoming rigid and unable to adapt later)
>
> **Extension to On-Policy (PPO):**
> Since PPO uses batches collected on-policy, we would compute the empirical FIM using the current batch of data used for the policy/value update. The main difference coefficient λ for on-policy dynamics.
>
> **Large Replay Buffers**
> FGSF does not inherently require large replay buffers; the FIM is computed on the mini-batch sampled for the standard update (see Q6b).
>
> **Savitzky-Golay Filter:**
> This filter was used only for visualization purposes. It's a standard technique in signal processing for estimating derivatives.
>
> **Pseudocode Clarification:**
> * a) Yes it is. We will clarify the notation.
> * b) N in the FIM equation represents the mini-batch size used for the update and FIM calculation
>
> **Small λ Values:**
> Yes, the small optimal values for λ are related to the large magnitude of the FIM trace. Normalizing inputs/outputs/rewards could potentially stabilize FIM calculations. This is an interesting avenue for future work.
>
> **Dormant Neurons (Fig 10):**
> Our current hypothesis (requiring further study) is that FGSF might retain neurons that become dormant but are still important for representing the learned function (potentially having high individual FIM contributions, thus receiving minimal perturbation from FGSF). Simply maximizing active neurons might not be optimal. This resonates with findings that network pruning can reduce PB susceptibility (Obando-Ceron et al., 2024).
>
> We hope these clarifications and our commitment to improving the paper warrant an increase in the score. Given space constraints, we welcome follow-up questions to address any remaining concerns in more detail.
> ### References
> See Reviewer 738E

---

> > ### Comment · Reviewer_LhZa · 2025-04-03
> >
> > I appreciate the authors' detailed response, which addresses many of my questions. I have accordingly increased the score. I hope the authors will incorporate the feedback to improve clarity. I also highly recommend that the authors have a plan to compare their approach with relevant methods used to address plasticity loss.
> >
> > > **Environments**  Would tasks like the bit-flipping problem (Dohare et al., 2024) be representative of what the reviewer has in mind?
> >
> > I didn't have a specific environment in mind. I was simply considering environments where your method would be well-suited. That said, both bit flipping and slippery Ant could be suitable. Tweaking existing DM Control environments with gradually changing settings could also be an option.
> >
> > I like this line of work and would like to see it succeed. I still have a couple of concerns and am willing to further increase my score if the authors can propose a reasonable plan to address the following:
> > 1. Evaluation on suitable non-stationary environments.
> > 2. Comparisons with methods proposed in plasticity loss research.

---

> > > ### Author Response · Authors · 2025-04-07
> > >
> > > We sincerely appreciate the reviewer's continued engagement and the raised score. We are fully committed to incorporating the feedback to enhance the clarity and impact of our work.
> > >
> > > ## **Evaluation**
> > > Regarding the evaluation on suitable non-stationary environments, we are actively exploring the bit-flipping problem. Its reduced computational complexity allows us to isolate and focus on the non-stationary aspects, minimizing potential confounding variables. We are already observing this to be a clear and compelling environment that demonstrates FGSF's capabilities.
> > >
> > > ## **Comparison**
> > > For the comparison with methods proposed in plasticity loss research, we are currently integrating these techniques into our experimental framework. We are particularly interested in understanding how Continual Backprop, ReDo, and Layer Norm + Weight Decay compare to FGSF. Our preliminary explorations suggest that each method addresses the plasticity challenge from distinct angles, and we are eager to quantify their respective strengths and weaknesses within our problem setting further.
> > >
> > > Specifically, we are in the process of strengthening and confirming some of the findings that already appeared in our Appendices:
> > >
> > > * **Continual Backprop:** We have kept observing that, in line with the results shown in our ablation study (Sec 5.3, Fig 7), an unstructured noise injection strategy can already offer benefits which are, however, different from Continual Backprop's structured noise injections. The latter noise injections also result in  differences with our FGSF's FIM-guided approach.
> > > * **ReDo:** Our further analysis of dormant neurons based on the results presented in Appendix B.2, Fig 10 keeps indicating a complex interplay between dormancy and performance. We are exploring how ReDo's approach to resetting dormant neurons contrasts with FGSF's nuanced handling of these neurons.
> > > * **Layer Norm + Weight Decay:** While this method represents a different approach, we are working to integrate it into our comparisons to better understand its performance relative to FGSF. Yet at the moment, we do not have any results for this.
> > >
> > > We understand the importance of these comparisons and are prioritizing their inclusion in our ongoing research. We are confident that these additional experiments will further solidify the contribution of FGSF and provide a comprehensive evaluation of its effectiveness. We hope that their integration within the paper will allow the reviewer to consider our contribution as a strong accept.
> > >
> > >
> > > We are grateful for the reviewer's encouragement and constructive feedback, and thank appreciate their engagement in the rebuttal process.

---

### Official Review · Reviewer_mZKQ · 2025-03-14

**Overall Recommendation:** 3

**Summary:**

This paper addresses the primacy bias (PB) problem in deep reinforcement learning, where early experiences disproportionately influence the learning process. The authors propose a novel method called Fisher-Guided Selective Forgetting (FGSF) that leverages the Fisher Information Matrix (FIM) to selectively modify network weights and prevent early experiences from dominating. The paper presents a characterization of PB through FIM trace patterns, identifying memorization and reorganization phases during learning. The authors evaluate FGSF across DeepMind Control Suite environments, analyze its impact on actor and critic networks, and examine the role of replay ratios and noise injection. Results show that FGSF consistently outperforms baseline methods, particularly in complex environments.

**Claims And Evidence:**

The paper's claims about FGSF's effectiveness are generally supported by empirical evidence across multiple environments. The authors show performance improvements compared to baseline SAC and network resetting approaches, particularly in complex environments like Humanoid and Quadruped. However, the evidence would be more convincing if the authors had compared FGSF with other state-of-the-art PB mitigation techniques beyond just network resetting. The characterization of PB through FIM trace patterns is interesting but would benefit from more rigorous statistical validation.

**Essential References Not Discussed:**

no

**Experimental Designs Or Analyses:**

The experimental design is generally sound, with appropriate baselines and ablation studies. The use of the FIM trace to characterize learning phases is novel and insightful. However, I noticed that the hyperparameter sensitivity analysis is somewhat limited, focusing primarily on the scrubbing coefficient λ while keeping the scrubbing frequency fixed at 10. A more comprehensive exploration of hyperparameter interactions would strengthen the analyses.

**Methods And Evaluation Criteria:**

The proposed methods are well-motivated and the evaluation using the DeepMind Control Suite is appropriate for assessing reinforcement learning algorithms. The ablation studies examining critic-only versus full network scrubbing provide valuable insights into the method's mechanics. However, the evaluation would be strengthened by including comparisons with other advanced PB mitigation techniques, such as plasticity injection methods and self-distillation approaches mentioned in the related work section.

**Other Comments Or Suggestions:**

The paper is generally well-written, but some figures (particularly Figures 3, 8, 11, and 12) contain too many subplots with small font sizes, making them difficult to interpret. The authors could consider reorganizing these figures or focusing on the most important environments for the main paper. Additionally, the explanation of the FIM approximation using EKFAC could be expanded to ensure reproducibility.

**Other Strengths And Weaknesses:**

Strengths include the novel perspective of using FIM for characterizing and addressing primacy bias, the extensive experimental evaluation across various environments, and the insightful ablation studies that uncover the differential impact on actor and critic networks.

**Questions For Authors:**

Why did you limit your comparisons to only network resetting as a baseline PB mitigation technique, rather than including more recent approaches like plasticity injection or self-distillation? Including such comparisons would provide a more comprehensive evaluation of FGSF's relative effectiveness.

**Relation To Broader Scientific Literature:**

The paper effectively connects ideas from information geometry, machine unlearning, and deep reinforcement learning. The authors appropriately contextualize their work within existing literature on primacy bias and acknowledge related approaches.

**Theoretical Claims:**

no

---

> ### Author Rebuttal · Authors · 2025-03-31
>
> We sincerely thank Reviewer mZKQ for their positive assessment and constructive feedback.
>
> ### Response to Comparison with other SOTA PB methods:
>
> We agree that comparisons with additional recent Primacy Bias (`PB`) mitigation techniques (e.g., specific plasticity injection, self-distillation) would further contextualize `FGSF`'s performance and represent valuable future work. For this initial study introducing our Fisher Information Matrix (`FIM`)-guided approach, we focused on comparing against baseline Soft Actor-Critic (`SAC`) and the foundational periodic reset method (Nikishin et al., 2022). This allowed us to clearly establish the distinct advantages (e.g., stability, performance in complex `PB`-prone tasks) offered by `FGSF`'s principled, geometrically-motivated mechanism compared to a fundamental heuristic baseline.
>
> ### Response to Hyperparameter Sensitivity:
>
> We focused our sensitivity analysis (Fig 4) on the scrubbing coefficient `λ`, which directly controls the intervention magnitude (Sec 3.3/3.4). We kept the scrubbing frequency `F` fixed because magnitude (`λ`) and frequency (`F`) are inherently coupled; higher frequency generally necessitates lower magnitude for stability. Isolating `λ` allowed a clearer and simpler analysis of its primary effect. While our results demonstrate robustness across a practical range of `λ` values, we agree that explicitly studying the interaction between `λ` and `F` is an important direction for future investigation.
>
> ### Response to Other Comments (Figure Readability and EKFAC Explanation):
>
> We thank the reviewer for the constructive suggestions regarding presentation.
>
> * **Figures**: We agree that Figures 3, 8, 11, and 12 are dense due to presenting results across multiple environments for completeness. In the camera-ready version, we will revise these figures to improve readability, potentially by increasing font sizes, focusing the main paper plots on the most illustrative environments (e.g., `Humanoid`, `Quadruped`, `Pendulum`), and moving others to the appendix if necessary.
> * **EKFAC**: We apologize for the brevity of the `EKFAC` explanation. We will expand this in the camera-ready version (likely in Sec 4 or Appendix), providing a clearer reference to the specific approximation used (George et al., 2021) and outlining the key implementation details relevant to `FGSF` to enhance reproducibility.
>
> We appreciate the reviewer's valuable feedback and positive evaluation. We hope these responses clarify our choices and the contributions of our work. Based on these clarifications, we hope the reviewer feels confident in maintaining or potentially strengthening their assessment. Given space constraints, we welcome follow-up questions to address any remaining concerns in more detail.
>
> ### References
>
> 1. George, Thomas (2021). “NNGeometry: easy and fast fisher information matrices and neural tangent kernels in PyTorch.”
> 2. Nikishin, Evgenii, Max Schwarzer, Pierluca D’Oro, Pierre-Luc Bacon, and Aaron Courville (2022). “The primacy bias in deep reinforcement learning.” In: International conference on machine learning. PMLR, pp. 16828–16847.

---

### Official Review · Reviewer_PPSq · 2025-03-15

**Overall Recommendation:** 2

**Summary:**

This paper proposed that the Fisher information can be a indicator on the plasticity loss during training. Based on the increasing and decreasing tendency of the trace of the Fisher information matrix, the authors proposed a novel method for tackling the plasticity loss by adding noise to the weights periodically. The scale of the noise is proportional to the inverse of the degree of the Fisher information, which indicates that the weights having low Fisher information value can be largely perturbed to promote the plasticity of the network. In the experiment, there are some cases that the proposed method outperforms the baseline (e.g. reset) in DMC environment.

**Claims And Evidence:**

The claims are not convincing since the empirical evidence for the relationship between FIM and the plasticity loss cannot explain the experiment results.

**Essential References Not Discussed:**

N/A

**Experimental Designs Or Analyses:**

The experimental desings are valid.

**Methods And Evaluation Criteria:**

The evaluation citeria does make sense.

**Other Comments Or Suggestions:**

Already mentioned in the above section.

**Other Strengths And Weaknesses:**

Strengths

1. Different from the claims in previous methods (e.g. ReDo [1], InFeR [2]), this paper pointed out that the degree of the Fisher information can ba a indicator for determining the plasticity loss.

Weaknesses

1. Compared to the basline (i.e. reset), the overall performance of the FGSF is somewhat incremental. Furthermore, in case of Acrobat task, applying FGSF rather hinders learning the task.

2. Threre is no experimental resutls when the replay ratio (RR) is high for other tasks. Since FGSF already achieves better performance in Quadruped task, it is not persuasive on showing only the results for Quadruped task.

3. It is confusing that the intuition behind using the FIM as the indicator on the plasticity loss is a proper way. In the Quadruped task, the FIM value is much lower than SAC, but the overall performance is much better than the SAC. How can we explain this result? As the FIM value is low, the sacle factor for perturbing the weight is high, which promotes the forgetting to prevent the plasticity loss. Therefore, if the FGSF is applied properly, the FIM value should be larger than that of SAC. Furthermore, in case of the Acrobot, the overall tendency is totally different from the Quadruped. Does the FGSF can be applied generally across the all tasks?

[1] Sokar et. al., The Dormant Neuron Phenomenon in Deep Reinforcement Learning, ICML, 2023

[2] Lyle et. al., Understanding and Preventing Capacity Loss in Reinforcement Learning, ICLR, 2022

**Questions For Authors:**

Already mentioned in the  Strengths And Weaknesses section

**Relation To Broader Scientific Literature:**

N/A

**Theoretical Claims:**

There is no theoretical claims.

---

> ### Author Rebuttal · Authors · 2025-03-31
>
> We thank Reviewer PPSq for their insightful feedback. We address the weaknesses and questions below.
> ### Weakness 1:
> While final reward gains may seem incremental in some tasks vs reset peak, FGSF offers significant advantages:
> * **Learning Stability**: Avoids reset's drastic periodic performance drops (Fig 2).
> * **Sample Efficiency**: Often faster convergence vs baseline SAC (e.g., 20-30% fewer steps in complex environments, Fig 2).
> * **Targeted Improvement**: Substantial gains in complex, high-PB environments (Humanoid, Quadruped, Table 1, Fig 2), aligning with our PB characterization (Sec 3.2).
>
> Regarding Acrobot failure: We acknowledge FGSF (and baseline/reset) struggled on Acrobot. We attribute this to shared hyperparameters from Haarnoja et al. (2018) (used for fair comparison), under which prior work also reported difficulties (e.g., Nikishin et al., 2022). This suggests task sensitivity or hyperparameter issues common to these methods, rather than a failure of the PB mitigation strategy itself, as PB (characterized by the two-phase FIM pattern) is less dominant in this simpler task.
> ### Weakness 2:
> Figure 5 focuses on Quadruped for high replay ratio (RR) analysis because our FIM trace analysis (Fig 1, 5.3, Sec 3.2) indicates it has one of the strongest Primacy Bias (PB) signatures (most pronounced 'Memorization phase' FIM spike for baseline SAC). This makes it the most relevant and challenging test case for evaluating PB mitigation where this phenomenon is amplified (high RR).
> Due to computational demands and the submission timeline, we focused resources on this representative case. Our internal preliminary results on other environments were consistent with RR=1 findings and the detailed Quadruped analysis: FGSF maintained better stability and performance compared to baseline SAC, particularly where the PB signature was originally evident. The core finding presented – that FGSF's advantage is amplified when PB is exacerbated by high RR – was thus robustly demonstrated in the most critical scenario and supported by initial findings elsewhere. We will add more results, confirming these insights, in the camera-ready version.
> ### Weakness 3:
> The reviewer raises excellent questions about interpreting the FIM trace dynamics and the generality of FGSF. This is a central part of our contribution and we welcome this opportunity to clarify our approach by explicitly linking this to our PB characterization framework (Sec 3.2).:
> * **FIM Trace and Performance (Quadruped)**: Our framework identifies PB via a two-phase FIM trace pattern: an initial, high-sensitivity 'Memorization Phase' (rapid Tr(F) increase) followed by a 'Reorganization Phase' (Tr(F) decrease). The first pahse, where the network rapidly overfits early data, is particularly problematic. In Quadruped, baseline SAC clearly exhibits this pathological Memorization Phase with a massive Tr(F) spike (Fig 3), correlating with poor final performance due to PB. FGSF's success comes from suppressing this detrimental Memorization Phase. The resulting lower Tr(F) observed in FGSF is evidence that it presents the extreme sensitivity and overfitting characteristic of early PB stages. This aligns with findings (Hochreiter & Schmidhuber 1997; Jastrzebski et al. 2021) linking lower early Tr(F) and flatter minima to better generalization. Thus, FGSF's controlled FIM trace leads to superior performance by avoiding the PB-induced pathological state.
>
> * **Adaptive Perturbation Mechanism**: The reviewer's intuition is correct: FGSF's $FIM^{-1/4}$ scaling means more perturbation is applied when Tr(F) is low (i.e., sensitivity is low). This is crucial for maintaining plasticity after FGSF has successfully navigated past (or prevented) the damaging Memorization Phase. It keeps the network adaptable without succumbing to early overfitting. We note that the goal isn't necessarily a higher final Tr(F) than the baseline, but rather achieving a healthy state via a controlled trajectory that avoids the initial Tr(F) explosion.
> * **Acrobot and Generality**: The different FIM dynamics in Acrobot (Fig 2/3) highlight that the PB signature itself (the strong two-phase pattern) is task-dependent. In simpler tasks, the intense Memorization Phase characteristic of PB might be absent or significantly weaker. As FGSF is designed to mitigate PB by intervening based on these FIM dynamics, it naturally has less impact when the PB signature it targets is not the primary factor of poor performance. The principle of FGSF is general, but its effectiveness as a PB mitigation tool directly correlates with the presence and strength of the PB phenomenon as characterized by our paper.
>
> We hope these clarifications address the points raised and demonstrate our work's value and novelty. We ask the reviewer to consider these clarifications in the final evaluation. Given space constraints, we welcome follow-up questions to address any remaining concerns in more detail.
> ### References
> See Reviewer 738E

---

> > ### Comment · Reviewer_PPSq · 2025-04-06
> >
> > Thank you for the authors' effort on the rebuttal comments.
> >
> > I carefully read the comments, and some notions on FGSF become clear.
> >
> > However, I wonder the contribution on FGSF is still strong. Though the authors explained on the intuition behind using the Fisher informantion for diagnosing the primacy bias, I think the overall tendency is still inconcsistent with the intuition.
> >
> > For the above reason, I will keep my score.

---

> > > ### Author Response · Authors · 2025-04-06
> > >
> > > We sincerely appreciate the reviewer for carefully reading our comments and taking the time to reassess our work.
> > >
> > > We kindly ask if it would be possible to further elaborate on the exact reasons, possibly with some examples, that make the reviewer believe that there is an inconsistent behavior between the intuition driving our FGSF approach and its empirical performance.

---

### Decision · Program_Chairs · 2025-05-01

**Decision:**

Reject

**Comment:**

This paper presents a mechanism for mitigating primacy bias in deep RL via the Fisher information matrix. The authors conduct a series of experiments to demonstrate the relationship between primacy bias and the Fisher information matrix, as well as the effectiveness of their method in mitigating the impact of the primacy bias phenomenon.

All reviewers found this work well-written, interesting, and the idea with merit. There are some concerns regarding the breadth of experimentation and whether it is sufficient to demonstrate this approach as a generalizable mechanism; some further concerns are regarding the sensitivity of hyper-parameters and whether this approach is robust to values outside the settings considered, in particular the influence of $\lambda$ and $F$ on each other.

Overall, I agree with most of the reviewers that this work is interesting and establishes a novel connection between the Fisher information matrix and primacy bias (and plasticity loss, by extension). However, the reviewers have raised a number of points and suggestions that would strengthen the impact of this work. The authors have acknowledged most of these and suggested they are "actively working on it" (see [response to LhZa](https://openreview.net/forum?id=uOGK8zwgvc&noteId=gMR116007O)).

Thus, I believe the current version is not quite ready for publication; the paper would be much stronger were the authors to incorporate the feedback received and include the additional experiments they are currently pursuing. I encourage the authors to integrate the feedback received during the review/discussion phase for a resubmission.